# Lateral septum adenosine $A_{2A}$ receptors control stress-induced depressive-like behaviors via signaling to the hypothalamus and habenula

Muran Wang[1,9], Peijun Li [2,3,9], Zewen Li[1,9], Beatriz S. da Silva[4,5], Wu Zheng[1], Zhenghua Xiang[6], Yan He[1], Tao Xu[1], Cristina Cordeiro[4,5], Lu Deng[2,3], Yuwei Dai[1], Mengqian Ye[1], Zhiqing Lin[1], Jianhong Zhou[1], Xuzhao Zhou[1], Fenfen Ye[1], Rodrigo A. Cunha [4,7], Jiangfan Chen [1,8] ✉ & Wei Guo [1] ✉

Major depressive disorder ranks as a major burden of disease worldwide, yet the current antidepressant medications are limited by frequent non-responsiveness and significant side effects. The lateral septum (LS) is thought to control of depression, however, the cellular and circuit substrates are largely unknown. Here, we identified a subpopulation of LS GABAergic adenosine $A_{2A}$ receptors ($A_{2A}$R)-positive neurons mediating depressive symptoms via direct projects to the lateral habenula (LHb) and the dorsomedial hypothalamus (DMH). Activation of $A_{2A}$R in the LS augmented the spiking frequency of $A_{2A}$R-positive neurons leading to a decreased activation of surrounding neurons and the bi-directional manipulation of LS-$A_{2A}$R activity demonstrated that LS-$A_{2A}$Rs are necessary and sufficient to trigger depressive phenotypes. Thus, the optogenetic modulation (stimulation or inhibition) of LS-$A_{2A}$R-positive neuronal activity or LS-$A_{2A}$R-positive neurons projection terminals to the LHb or DMH, phenocopied depressive behaviors. Moreover, $A_{2A}$R are upregulated in the LS in two male mouse models of repeated stress-induced depression. This identification that aberrantly increased $A_{2A}$R signaling in the LS is a critical upstream regulator of repeated stress-induced depressive-like behaviors provides a neurophysiological and circuit-based justification of the antidepressant potential of $A_{2A}$R antagonists, prompting their clinical translation.

Major depressive disorder ranks as the first cause of the burden of disease, affecting more than 300 million people worldwide[1]. A major risk of depression is suicide ideation, with over 65% of individuals that died by suicide being affected by mood disorders[2]. The traditional antidepressant treatments, such as with selective serotonin reuptake inhibitors, are limited by the slow onset of therapeutic effects, their limited effect size and frequent non-responsiveness whereas newly emerging fast-acting antidepressants, such as ketamine, are associated with significant side effects such as addiction propensity[3]. Thus, the identification of novel and effective therapeutic targets to develop antidepressants is critically needed.

Convergent epidemiological, genetic, and pharmacological findings support the role of adenosine $A_{2A}$ receptors ($A_{2A}$R), an important neuromodulatory receptor[4,5], as a novel therapeutic target for depression. Human epidemiological investigation into dietary factors associated with depression unveiled that the consumption of caffeine,

the most widely consumed psychoactive drug, by acting as a non-selective adenosine receptor antagonist[6,7], is inversely correlated with depression and the likelihood of suicide in several cohorts (reviewed in[8,9]). Also, polymorphisms of the human $A_{2A}R$ gene are associated with the incidence and clinical heterogeneity of depression[10]. This potential link of adenosine signaling with depression is attractive since depression is intrinsically linked to chronic stress, adenosine levels are increased upon brain stressful conditions (reviewed in[11]) and the upregulation of $A_{2A}R$ in different brain regions is a proposed bio-marker of the onset of neuropsychiatric diseases[12]. The stress-induced alterations of the adenosine neuromodulation system are initially homeostatic in nature, but their persistent alteration upon chronic stress results in synaptic dysfunction[11], which has been proposed to underlie depressive-like behaviors[13]. Indeed, repeated restraint stress[14] and chronic unpredictable stress[15] induce the upregulation of synaptic $A_{2A}R$, and the genetic deletion or pharmacological blockade of $A_{2A}R$ attenuate maladaptive features in various depressive-like behavioral paradigms[14–19]. Notably, the antidepressant-like effect size of $A_{2A}R$ antagonists is equivalent to that of classical antidepressants such as desipramine in the rat learned helplessness model[18]. Although the use of forebrain $A_{2A}R$ knockout mice has pinpointed a role for central $A_{2A}R$ in the control of depressive-like phenotypes[15], $A_{2A}R$ are located in various types of neurons in brain areas that are associated with depression[20], including medial prefrontal cortex (mPFC), amygdala, and hippocampus. So, identifying the critical locus and neural circuits for $A_{2A}R$ antagonism to elicit antidepressant activity is still a key hurdle for the acceptance and development of $A_{2A}R$-based therapies.

Recently, we have found the expression of $A_{2A}R$ in the lateral septum (LS) via a novel knock-in transgenic mouse line ($A_{2A}R$-tag mice)[21]. The LS, a midline brain structure, has been implicated in a wide variety of functions, such as emotional, motivational, and social behavior[22,23]. Evidence is accumulating to relate an abnormal function of the LS with a directional control of stress-induced depressive behavior (reviewed in[22,24]): thus, alterations of neural activity within the LS correlate with the development of behavioral manifestations of depression in animal models, antidepressant drugs affect LS functioning and cell signaling processes in LS neurons are associated with antidepressant-like effects. Recent studies have identified that LS-GABAergic neurons trigger depression-related behaviors through their periaqueductal gray projections[25] and somatostatin-positive neurons in LS can also influence depression-like behaviors[26]. However, disentangling the precise LS circuitry associated with depressive phenotypes is still far from clear, since the LS is constituted by a majority of GABAergic neurons forming multiple superimposable circuits inter-linked by intra-septal intrinsic sub-circuits, so that LS sub-circuits often antagonize each other to influence behavioral outputs[22,27]. LS circuits display a complex architecture of gene expression, especially of modulatory receptors[27], such as $A_{2A}R$, which adds further complexity to unraveling the LS circuitry controlling mood dysfunction.

Our present study identifies a subpopulation of LS GABAergic $A_{2A}R$-positive neurons mediating depressive symptoms via direct projects to the dorsomedial hypothalamus (DMH) and the lateral habenula (LHb). Thus, the selective increase of $A_{2A}R$ activity in the LS, shown in two chronic stress-induced depression mouse models, was necessary and sufficient for the development of depressive-like behaviors. Given the noted clinical safety profile of $A_{2A}R$ antagonists[28], these insights offer an opportunity to manage depressive disorders by targeting LS-$A_{2A}R$ signaling.

## Results

### $A_{2A}R$ expression identifies a subset of GABAergic LS neurons

Our previous study has shown that $A_{2A}R$-positive ($A_{2A}R^+$) neurons make up about one percent of neurons in the LS, as shown by anti-HA staining in $A_{2A}R$-tag mice[29]. To identify the projection areas of LS-$A_{2A}R^+$ neurons, rAAV2/9-hSyn-DIO-EYFP were injected into the LS of $A_{2A}R$-Cre

mice. Three weeks after injecting, the EYFP-positive axonal fibers (i.e. from LS-$A_{2A}R^+$ neurons) were only found in the DMH and the LHb (Fig. 1a). No such positive-EYFP signal (Supplementary Fig. 1) was found in periaqueductal gray (PAG), ventral tegmental area (VTA) and raphe which also receive projections from LS as reported in previous studies[22,24]. To further validate the direct connection between LS-$A_{2A}R^+$ neurons and DMH or LHb, we injected the green retrograde tracer recombinant cholera toxin subunit B (CTB-488) into the DMH or LHb of $A_{2A}R$-tag mice (Fig. 1b). Five days later, we observed a co-location of CTB and HA fluorescence in the LS, confirming the direct projections from the LS-$A_{2A}R^+$ neurons to the LHb and DMH (Fig. 1c, d).

To explore the cellular effect of the $A_{2A}R$ signaling in the LS, we determined the effect of the $A_{2A}R$ agonist CGS21680 on neuronal firing of LS-$A_{2A}R^+$ neurons by in vitro electrophysiological recordings in acute brain slices from adult $A_{2A}R$-Cre mice injected with rAAV2/9-hSyn-DIO-EYFP into the LS. The activity of EYFP-positive cells (i.e. LS-$A_{2A}R^+$ neurons) was recorded using a cell-attached voltage-clamp mode in low-$Mg^{2+}$ (1 mM) aCSF solution. After cell-attached recording, we applied negative pressure to form a gigaohm seal between the cell and the glass pipette, and then used a brief suction to break into the cell. Then the cell will be kept under whole cell voltage clamp mode for at least 10 minutes to allow the biocytin (1%, Thermo Scientific) in the intracellular solution to diffuse well into the cell. This allowed a post-hoc morphological characterization of the recorded EYFP-positive neurons, which showed that $A_{2A}R^+$ neurons possessed long and numerous branches (Fig. 2a). A subsequent immunofluorescent staining of slices revealed that dorsal LS-$A_{2A}R^+$ neurons were GABAergic neurons (Fig. 2b), consistent with a previous study showing that the dorsal LS is made up of only inhibitory neurons[30]. Additionally, RNAscope in situ hybridization analysis confirmed the co-location of $A_{2A}R$ and GAD65 + 67 in dorsal LS (Supplementary Fig. 2). Moreover, single cell RT-PCR analysis of total mRNA isolated from individual recorded cells also confirmed that all recorded EYFP-positive cells were GABAergic neurons (Fig. 2c). Activation of $A_{2A}R$ by CGS21680 (30 nM) significantly increased the firing frequency of EYFP-positive neurons (Fig. 2d), indicating that $A_{2A}R$ activation augments neuronal activity of LS-$A_{2A}R^+$ neurons. The effect of CGS21680 was reversible after washout as shown in all three recorded cells (Fig. 2e).

Previous investigations have demonstrated that the systemic or local administration of diverse antidepressants increases the neuronal firing rate of the LS, whereas some stressful situations decrease its firing rate[24]. To determine the effects of $A_{2A}R$ on the LS circuitry, we first determined the effect of CGS21680 on non-$A_{2A}R^+$ neurons in LS, presumably also GABAergic neurons as most LS neurons[30], by in vitro electrophysiological recordings in brain slices of $A_{2A}R$-Cre mice injected with rAAV2/9-hSyn-DIO-EYFP. We found that $A_{2A}R$ activation by CGS21680 (30 nM) increased the frequency (but not amplitude) of spontaneous inhibitory postsynaptic currents (sIPSCs) of neurons around EYFP-positive cells, which likely resulted from the increase of $A_{2A}R^+$ GABAergic inputs to non-labeled neurons in the LS (Fig. 2f), although this remains to be experimentally confirmed since indirect effects, circuit-mediated or involving astrocytes may eventually be also involved.

To next determine the effects of $A_{2A}R$ on the activity of the LS as a whole, we measured c-Fos expression in the LS of C57BL/6 J mice after the focal infusion in the LS of either CGS21680 (5 μg/μL, 2 μL per injection) or vehicle (Fig. 3a). The focal microinjection of CGS21680 into the LS decreased the number of c-Fos-positive neurons in the LS compared to vehicle-treated mice (Fig. 3b). Furthermore, we found that CGS21680 infusion into the LS induced a robust upregulation of c-Fos expression in the LHb and DMH compared to vehicle-treated mice (Fig. 3c). Alterations found in other sub-regions in the hypothalamus (such as arcuate hypothalamic and ventromedial hypothalamic nucleus) may be an indirect effect considering the complex intrinsic circuitry of sub-regions of the hypothalamus (Supplementary Fig. 3).

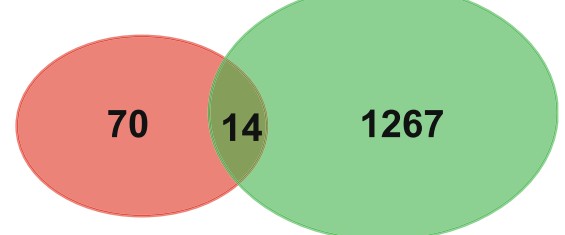

Fig. 1 | Dorsomedial hypothalamus (DMH) and lateral habenula (LHb) are the main outputs of the lateral septum (LS). a Confocal images of coronal sections showing LS $A_{2A}R^+$ neurons terminals in the DMH and LHb. Three weeks after injecting rAAV2/9-hSyn-DIO-EYFP into the LS of $A_{2A}R$-Cre mice, EYFP-positive axonal fibers were found in the DMH and LHb. Scale bar: 500/200 μm. LV, lateral ventricle; MS, medial septum. b Experimental scheme showing that CTB488 was injected into DMH or LHb of $A_{2A}R$-tag mice (to reveal $A_{2A}R$ expression via anti-HA staining). c Fluorescence images illustrating CTB488 (green) targeted to the DMH or LHb of $A_{2A}R$-tag mice and showing the co-location of DMH or LHb-projection neurons (green) and $A_{2A}R$ (red) in LS. Scale bars: 200/50 μm. 3 V, third Ventricle; MHb, medial habenula. d Average number of LS-$A_{2A}R^+$ neurons labeled with CTB488 from DMH or LHb (n = 3 mice/group, 6 slices/mouse). DMH-projecting LS-$A_{2A}R^+$ neurons, n = 32 ± 3 cells (yellow overlap); LHb-projecting LS-$A_{2A}R^+$ neurons, n = 14 ± 2 cells (yellow overlap). Source data are provided as a Source Data file.

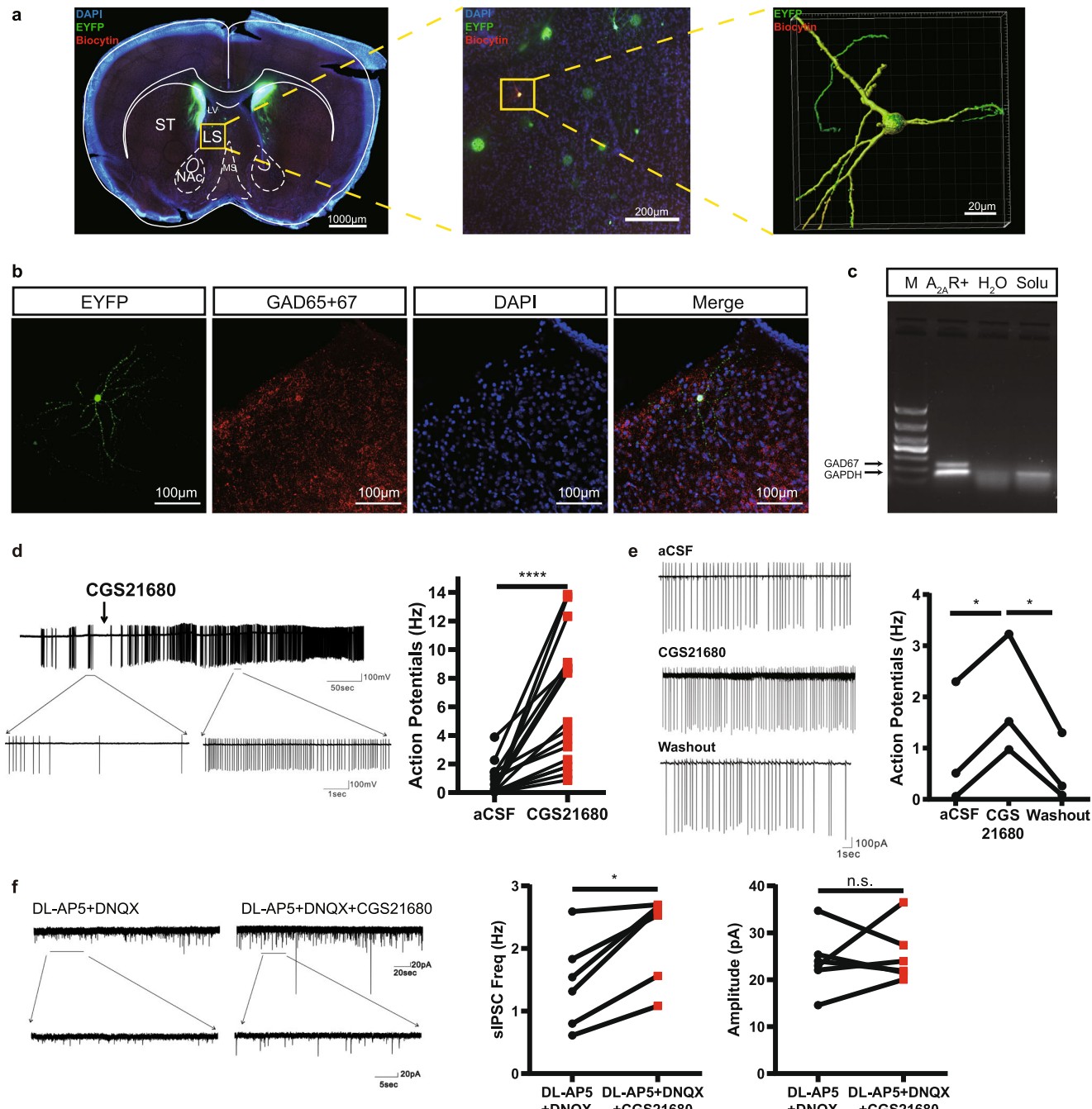

**Fig. 2 | Activation of $A_{2A}R$ in the lateral septum (LS) augments spiking frequency of LS-$A_{2A}R^+$ neurons with suppression of surrounding neurons.**
**a** Biocytin (red) in the intracellular solution diffused into the cells on brain slices from $A_{2A}R$-Cre mice injected with rAAV2/9-hSyn-DIO-EYFP into LS during the in vitro electrophysiological recordings. The recorded $A_{2A}R^+$ neurons (stained yellow) displayed long and numerous branches observed using confocal microscopy combined with 3D surface volume rendering. Scale bars: 1000/200/20 μm. LV, lateral ventricle; MS, medial septum; NAc, Nucleus Accumbens core; ST, striatum. **b** Representative image from a 300 μm slice after in vitro electrophysiology recording showing $A_{2A}R^+$ neurons (green) co-immunostained with anti-GAD65 + 67 antibody (red) in the LS. Nuclei are stained with DAPI in blue. Scale bar: 100 μm. **c** Single cell RT-PCR analysis with total mRNA isolated from recorded cells showed that all EYFP-positive cells were GABAergic neurons. M, 1000 bp marker; $A_{2A}R^+$, cell cytoplasm of LS-$A_{2A}R^+$ neuron; $H_2O$, sterile double distilled water; solu, pipette solution. **d** Representative trace and statistical graph showing that the activation of

$A_{2A}R$ by CGS21680 (30 nM) increased the firing frequency of EYFP-positive neurons ($n = 16$ cells from 8 mice, Wilcoxon test, $p = 0.00003$, W = 136.0). **e** Representative trace and statistical graph showing that the effect of CGS21680 on EYFP-positive neurons was reversible after washout (Ctrl: CGS21680, $n = 3$ cells from 3 mice, Paired t test, $p = 0.0309$, t(2)=6.663; CGS21680: washout, $n = 3$ cells, 3 mice, Paired t test, $p = 0.0238$, t(2) = 6.364). **f** Left: Representative voltage-clamp recording showing the alterations of spontaneous inhibitory post-synaptic currents (sIPSC, recorded upon blockade of glutamatergic activity with 50 μM DL-$AP_5$ and 20 μM DNQX) in LS non-$A_{2A}R^+$ neurons surrounding $A_{2A}R^+$ neurons before and after the application of CGS21680 (30 nM). Right: Statistical graph showing that CGS21680 increased sIPSCs frequency ($n = 6$ cells from 4 mice, Paired Wilcoxon test, $p = 0.0313$, W = 21), without altering their amplitude ($n = 6$ cells from 4 mice, Paired t test, $p = 0.6927$, W = 21). DL-$AP_5$, DL-2-amino-5-phosphonopentanoic acid; DNQX, 6,7-dinitroquinoxaline-2,3-dione. *$p < 0.05$, ****$p < 0.0001$; n.s., no significant difference. Source data are provided as a Source Data file.

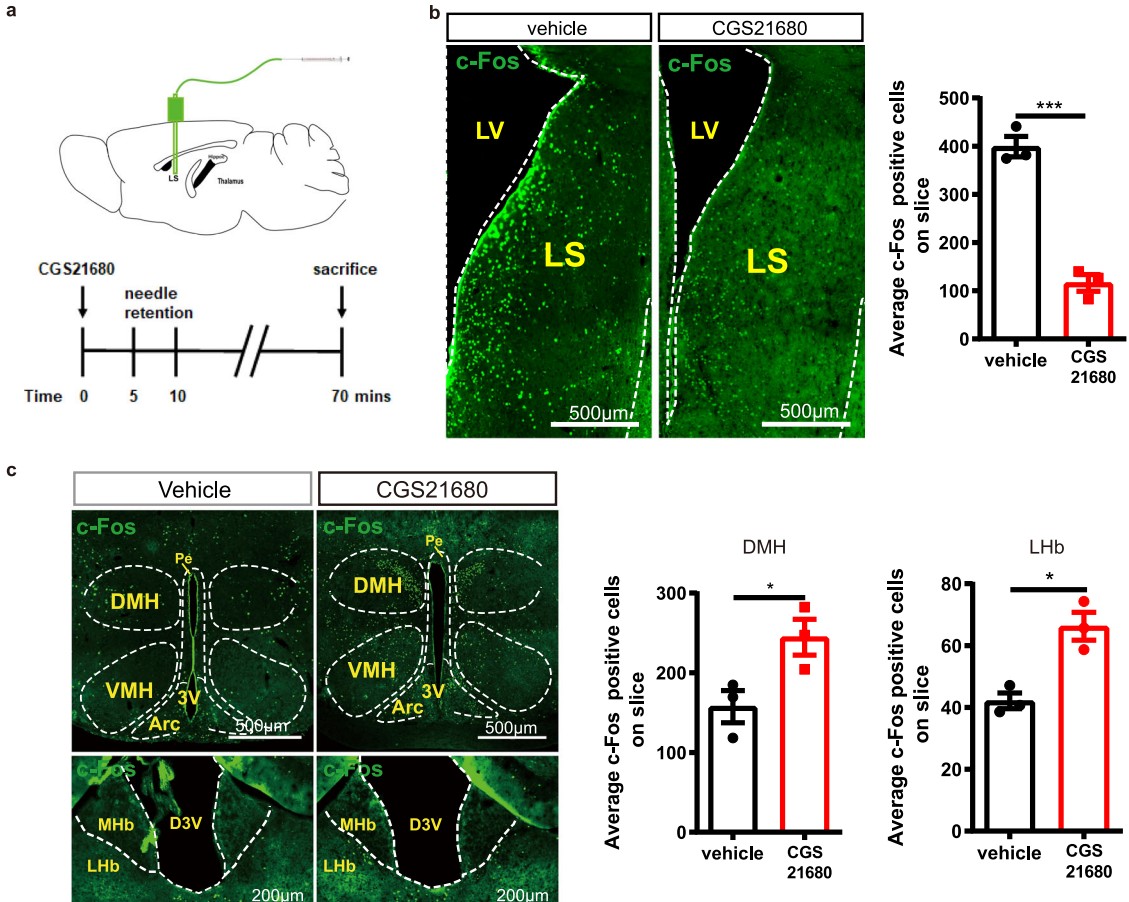

**Fig. 3 | The injection of the A$_{2A}$R agonist CGS21680 into the lateral septum (LS) suppresses c-Fos expression in the LS and increases c-Fos expression in the lateral habenula (LHb) and dorsomedial hypothalamus (DMH). a** Schematic protocol to test the role of the A$_{2A}$R agonist CGS21680 on LS outputs. **b** Representative immunofluorescence images and average bar graph illustrating the decreased expression of c-Fos in the LS ($n$ = 3 mice/group, 9 slices/mouse, Unpaired t test, $p$ = 0.0005, t(4) = 10.38). Scale bar: 500 μm. LV, lateral ventricle. **c** Representative immunofluorescence images and average bar graphs illustrating

the increased expression of c-Fos in the DMH ($n$ = 3 mice/group, 9 slices/mouse, Unpaired t test, $p$ = 0.0448, t(4) = 2.884) and LHb ($n$ = 3 mice/group, 9 slices/mouse, Unpaired t test, $p$ = 0.0075, t(4) = 4.648) after focal microinjection of the A$_{2A}$R agonist CGS21680 into the LS. Scale bars: 500/200 μm. Arc, arcuate hypothalamic nucleus; D3V, dorsal third ventricle; MHb, medial habenula; VMH, ventromedial hypothalamic nucleus; 3V, third Ventricle, Pe, periventricular hypothalamic. Data were shown as mean ± SEM. *$p$ < 0.05, ***$p$ < 0.001; n.s., no significant difference. Source data are provided as a Source Data file.

No change of c-Fos expression was found in the amygdala, PAG and VTA (Supplementary Fig. 4).

## Optogenetic modulation of the activity of LS-A$_{2A}$R$^+$ neurons formats depressive-like phenotype

We next optogenetically manipulated the activity of LS-A$_{2A}$R$^+$ neurons to determine their selective role in the regulation of depressive behavior. We bilaterally injected the rAAV2/9-Ef1α(H134R)-DIO-ChR2-EYFP or its control virus rAAV2/9-Ef1α-DIO-EYFP (200 nL) into the LS of male A$_{2A}$R-Cre mice and bilaterally implanted the optical fibers in the LS (Fig. 4a). Three weeks after viral expression, optogenetic activation of LS-A$_{2A}$R$^+$ neurons increased the immobility time in the tail suspension test (TST) compared to the control virus group (Fig. 4b), without affecting either the total traveled distance and the time spent in the center area in the open field test (OFT) (Fig. 4c, d) or the time spent in the open arms in the elevated O-maze (Fig. 4e). Additionally, we observed a similar depressive-like phenotype in female A$_{2A}$R-Cre mice after optogenetic activation of LS-A$_{2A}$R$^+$ neurons (Supplementary Fig. 5). Conversely, the selective inhibition of LS-A$_{2A}$R$^+$ neurons yield opposite behavioral effects compared with excitation. We bilaterally injected the rAAV2/9-Ef1α-DIO-eNpHR3.0-EYFP as well as control virus rAAV2/9-Ef1α-DIO-EYFP (200 nL) into

the LS of male A$_{2A}$R-Cre mice and bilaterally implanted the optical fibers in the LS (Fig. 4f). Three weeks after viral expression, optogenetic inhibition of LS-A$_{2A}$R$^+$ neurons decreased the immobility time in the TST compared to the control virus group (Fig. 3g) and did not modify behavior in both OFT and elevated O-maze test (Fig. 4h–j).

## Optogenetic modulation of the LS-A$_{2A}$R$^+$ → DMH or LS-A$_{2A}$R$^+$ → LHb projections influences depressive-like behaviors

The identification of the functional connectivity of the LS-A$_{2A}$R$^+$ → DMH and LS-A$_{2A}$R$^+$ → LHb, together with the important role of the hypothalamus and habenula in the control of mood behaviors[31,32], led us to investigate if the optogenetic activation of LS → LHb projections and of LS → DMH projections would reveal their putative critical role in the induction of depressive-like behaviors. We achieved the targeted expression of ChR2 in A$_{2A}$R$^+$ projection terminals by bilaterally injecting the rAAV2/9-Ef1α-DIO-ChR2(H134R)-EYFP or its control virus rAAV2/9-Ef1α-DIO-EYFP (200 nL) into the LS of male A$_{2A}$R-Cre mice. We implanted the optical fibers unilaterally in the DMH or bilaterally in the LHb (Fig. 5a, e). Optogenetic activation of the LS-A$_{2A}$R$^+$ → DMH projections increased the immobility time in the TST compared to the control virus group (Fig. 5b). There was no difference in the total traveled distance or in the time spent in the central area in

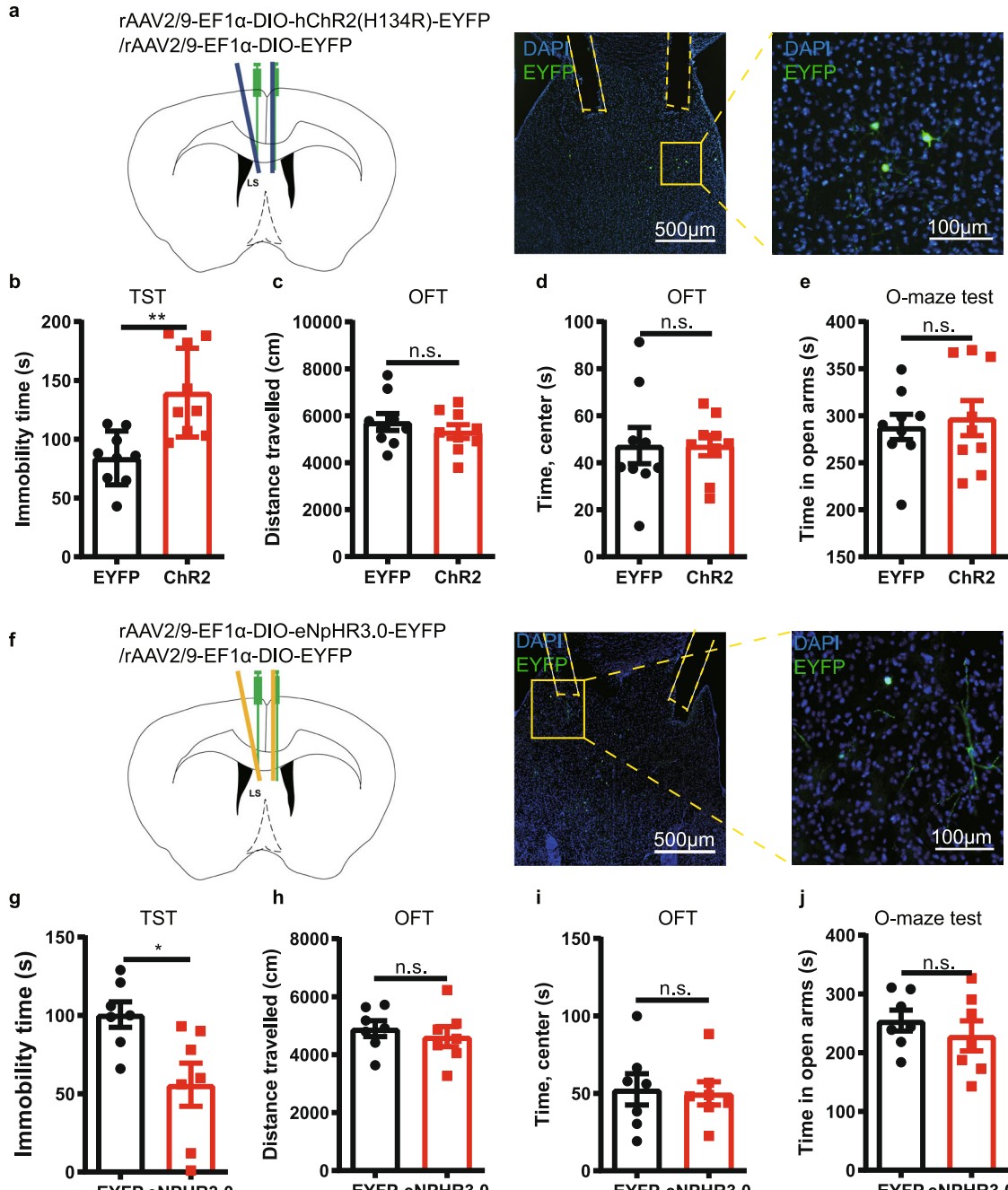

**Fig. 4 | Optogenetic modulation of the activity of A$_{2A}$R$^+$ neurons in the lateral septum (LS) influences depressive-like phenotype. a** Left: Schematic illustration of the location of virus injection and optic fibers implantation in A$_{2A}$R-Cre mice. Right: A representative fluorescent image showing the ChR2-positive neurons (green) and the localization of the optic fibers. Nuclei are stained with DAPI in blue. Scale bar: 500/100 μm. (b-e) Optogenetic activation of LS-A$_{2A}$R$^+$ neurons increased the immobility time in the tail suspension test (TST) (*n* = 9 mice/group, Unpaired t test, *p* = 0.0017, t(16) = 3.7710) **b** without affecting the total distance traveled (*n* = 9 mice/group, Unpaired t test, *p* = 0.3980, t(16) = 0.8684) **c** or the time in the central area in the open field test (OFT) (*n* = 9 mice/group, Unpaired t test, *p* = 0.9980, t(16) = 0.002512) **d** and in the elevated O-maze test (*n* = 9 mice/group, Unpaired t test, *p* = 0.6868, t(16) = 0.4106) **e**. **f** Left: Schematic illustration of the

location of virus injection and optic fibers implantation in A$_{2A}$R-Cre mice. Right: A representative fluorescent image shows the eNpHR3.0 positive neurons (green) and the localization of the optic fibers. Nuclei are stained with DAPI in blue. Scale bars: 500/100 μm. Optogenetic suppression of the activity of LS A$_{2A}$R$^+$ neurons decrease the immobility time in the TST (*n* = 7 mice/group, Unpaired t test, *p* = 0.0159, t(12) = 2.804) **(g)**, without affecting the total movement distance in the OFT (*n* = 7 mice/group, Unpaired t test, *p* = 0.5465, t(12) = 0.6205) **(h)**, the time spending in the central area in the OFT(*n* = 7 mice/group, Unpaired t test, *p* = 0.8404, t(12) = 0.2058) **(i)**, and the duration in the open arm of the elevated O-maze **(j)** (*n* = 7 mice/group, Unpaired t test, *p* = 0.4188, t(12) = 0.8373). Data were shown as mean ± SEM. \**p* < 0.05, \*\**p* < 0.01; n.s., no significant difference. Source data are provided as a Source Data file.

the OFT between the ChR2 group and the control group (Fig. 5c, d). Similarly, optogenetic activation of the LS-A$_{2A}$R$^+$ → LHb projections increased the immobility time in the TST (Fig. 5f), without affecting the total distance traveled or the time spent in the central area of the OFT (Fig. 5g, h). Thus, optogenetic activation of the projection terminals

from the LS-A$_{2A}$R$^+$ neurons to the DMH or to the LHb reproduced the induction of depressive-like behaviors, with a more robust effect of LS-A$_{2A}$R$^+$ → DMH projections in the TST, in accordance with the larger number of projections from LS-A$_{2A}$R$^+$ neurons to the DMH compared to the LHb (Fig. 1a).

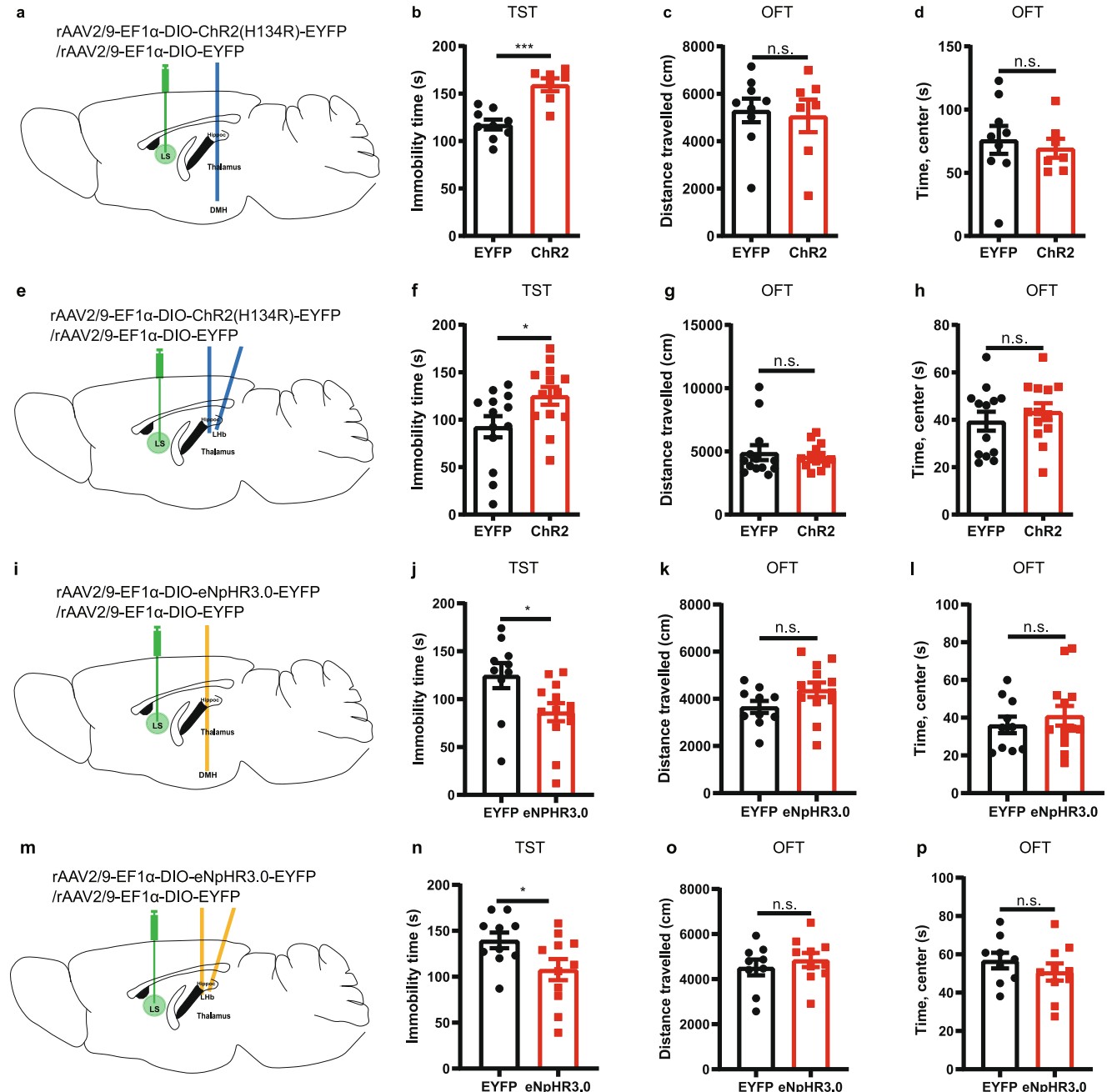

**Fig. 5 | Optogenetic modulation of the projection terminals in the dorsomedial hypothalamus (DMH) and lateral habenula (LHb) of A$_{2A}$R-containing neurons in the lateral septum (LS) influences depressive-like behaviors. a–d** Schematic representation of the optogenetic activation of the LS-A$_{2A}$R$^+$ → DMH terminals (**a**). Optogenetic activation of LS-DMH projection terminals of A$_{2A}$R-containing neurons increased immobility in the tail suspension test (TST) ($n = 9$ and 7 mice per group for EYFP and ChR2, respectively, Unpaired t test, $p = 0.0002$, t(14)=5.041) (**b**), without altering the total distance traveled ($n = 9$ and 7 mice per group for EYFP and ChR2, respectively, Unpaired t test, $p = 0.7847$, t(14)=0.2785) and the time spent in the center ($n = 9$ and 7 mice per group for EYFP and ChR2, respectively, Unpaired t test, $p = 0.6503$, t(14)=0.4633) in the open field test (OFT) (**c**, **d**). **e–h** Schematic representation of the optogenetic activation of the LS-A$_{2A}$R$^+$ →LHb terminals (**e**). Optogenetic activation of LS-LHb projection terminals increased immobility in TST ($n = 13$ mice/group, Mann-Whitney test, $p = 0.0441$, U = 45) (**f**), without altering the total distance traveled ($n = 13$ mice/group, Mann–Whitney test, $p = 0.5788$, U = 73) and the time spent in the center ($n = 13$ mice/group, Mann–Whitney test, $p = 0.4714$, U = 70) in the OFT (**g**, **h**). **i–l** Schematic representation of the optogenetic inhibition

of the LS-A$_{2A}$R-DMH terminals (**i**). Optogenetic inhibition of the LS-A$_{2A}$R$^+$ →DMH projections reduced the immobility time in the TST ($n = 10$ and 13 per group for EYFP and eNpHR3.0, respectively, Unpaired t test, $p = 0.0238$, t(21) = 2.437) (**j**), without altering the total distance traveled ($n = 10$ and 13 per group for EYFP and eNpHR3.0, respectively, Unpaired t test, $p = 0.0947$, t(21) = 1.750) and the time spent in the center ($n = 10$ and 13 per group for EYFP and eNpHR3.0, respectively, Unpaired t test, $p = 0.5013$, t(21) = 0.6843) in the OFT (**k**, **l**). **m–p** Schematic representation of the optogenetic inhibition of the LS-A$_{2A}$R-LHb terminals (**m**). Optogenetic inhibition of the LS-A$_{2A}$R$^+$ →LHb projections reduced the immobility time in the TST (10 and 11 per group for EYFP and eNpHR3.0, respectively, Unpaired t test, $p = 0.0418$, t(19) = 2.183) (**n**), without affecting the total distance traveled ($n = 10$ and 11 per group for EYFP and eNpHR3.0, respectively, Unpaired t test, $p = 0.4878$, t(17) = 0.7093) or the time spent in the central area of the OFT ($n = 10$ and 11 per group for EYFP and eNpHR3.0, respectively, Unpaired t test, $p = 0.3394$, t(17) = 0.9830) (**o**, **p**). Data were shown as mean ± SEM. *$p < 0.05$, ***$p < 0.001$; n.s., no significant difference. Source data are provided as a Source Data file.

To rule out the possibility that LS $A_{2A}R^+$ collateral projections to brain regions other than LHb and DMH might be responsible for the observed behavioral effects, optogenetic inhibition of LHb and DMH terminals was carried out. We achieved the targeted expression of eNpHR3.0 in $A_{2A}R^+$ projection terminals by injecting bilaterally the rAAV2/9-Ef1α-DIO-eNpHR3.0-EYFP or its control virus rAAV2/9-Ef1α-DIO-EYFP (200 nL) into the LS of male $A_{2A}R$-Cre mice. We implanted the optical fibers in the DMH or LHb (Fig. 5i, m). Optogenetic inhibition of the LS-$A_{2A}R^+$ → DMH projections reduced the immobility time in the TST compared to the control virus group (Fig. 5j). There was no difference in the total traveled distance or in the time spent in the central area in the OFT between the eNpHR3.0 group and the control group (Fig. 5k, l). Similarly, optogenetic inhibition of the LS-$A_{2A}R^+$ → LHb projections reduced the immobility time in the TST (Fig. 5n), without affecting the total distance traveled or the time spent in the central area of the OFT (Fig. 5o, p).

Together with the viral tracing results, these findings prompt the conclusion that the LS-$A_{2A}R^+$ → DMH and the LS-$A_{2A}R^+$ → LHb pathways mediate the depressive-like behavior caused by abnormal $A_{2A}R$ overactivity in the LS.

## $A_{2A}Rs$ are upregulated selectively in the LS in two mouse models of chronic stress

Since the upregulation of $A_{2A}R$ is a proposed biomarker of the dysfunctional brain circuits at the onset of brain diseases[12], it is relevant to test if $A_{2A}R$ is an inducible factor triggered by stress conditions. Thus, we quantified alterations of $A_{2A}R$ density in multiple brain regions associated with mood processing in two mouse models (male C57BL/6J mice) of chronic stress. First, after chronic restraint stress (CRS) exposure for 14 days, CRS-mice showed an increase in immobility time in the TST, without change of locomotion in the OFT, and of the time spent in the open arms in the elevated O-maze test (Fig. 6a–d). After behavioral testing, mice were immediately sacrificed and $A_{2A}R$ immunodensity was probed in several mood-associated brain regions, including the septum, prefrontal cortex (PFC), hippocampus, and striatum (Fig. 6e, f). Compared to the control group, mice subjected to CRS displayed a selective upregulation (1.82 times of control, $p = 0.005$) of $A_{2A}R$ density in the septum. Although $A_{2A}R$ immunodensity varied considerably across samples, we observed a trend for $A_{2A}R$ up-regulation also in the PFC, whereas the levels of $A_{2A}R$ remained unchanged in the striatum and hippocampus, consisting with previous Western blotting analyzes[33,34].

We also evaluated the regional pattern of $A_{2A}R$ upregulation in a second stress paradigm, the 5-day repeated forced swim stress (5d-RFSS) paradigm, described in a previous study[35]. Compared to control mice, stressed mice showed an increased immobility time in the forced swimming test (FST), without changes in the total distance of movement in the OFT (Supplementary Fig. 6a, b). 5d-RFSS-mice spent less time in the center area of OFT but similar time in the open arms of an elevated O-maze test compared to the control group (Supplementary Fig. 6c, d), consistent with previous reports [29]. After behavioral testing, the analysis of $A_{2A}R$ levels in the septum, PFC, hippocampus, striatum and hypothalamus revealed a selective increase (1.58 times of control, $p = 0.0003$) of $A_{2A}R$ density in the septum without significant changes in the three other brain regions (Supplementary Fig. 6e, f), as previously observed in the CRS model.

## $A_{2A}R$ overexpression in the LS is sufficient to induce a depression-like phenotype

We next investigated whether the $A_{2A}R$ overexpression in the LS was sufficient to trigger a depressive-like behavior. We confirmed achieving an $A_{2A}R$ overexpression in the LS of male C57BL/6J mice by an unilateral injection of $A_{2A}R$-expressing virus (AAV2/9-hSyn-$A_{2A}R$-3xflag-ZsGreen) but not of control virus (AAV2/9-hSyn-ZsGreen) (Fig. 7a, b). Three weeks after the viral injection, behavioral analysis

revealed that mice transfected with $A_{2A}R$-expressing virus displayed increased immobility in the TST (Fig. 7c). There was no difference in the total traveled distance and in the time spent in the center area in the OFT (Fig. 7d, e) nor in the time spent in the open arms in the elevated O-maze (Fig. 7f), between mice injected with the $A_{2A}R$-expressing virus and with the control virus. Furthermore, LS-$A_{2A}R$ overexpression also decreased the consumption of sucrose compared with the control group (Fig. 7h) and showed no difference in the total consumption of liquid (Fig. 7g). Thus, $A_{2A}R$ overexpression in the LS was sufficient to induce depressive-like behavior, indicating that $A_{2A}R$ upregulation in the LS is a modulator rather than only a biological marker of depressive-like behavior.

## Focal genetic and pharmacological $A_{2A}R$ inactivation selectively in the LS attenuate depressive-like phenotype

To further test the function of endogenous $A_{2A}R$ activity in vivo, we determined the effect of shRNA knockdown of LS-$A_{2A}R$ on depressive-like behaviors. We achieved a focal knockdown of LS-$A_{2A}R$ by unilaterally injecting into the LS of male C57BL/6J mice of AAV2/9-syn-$A_{2A}$RshRNA-GFP ($A_{2A}R$-shRNA) but not the control virus (AAV2/9-syn-$A_{2A}$Rshcontrol-GFP, ctrl-shRNA) (Fig. 8a). Focal knockdown of LS-$A_{2A}R$ reduced immobility in the TST (Fig. 8b) without affecting spontaneous motor activity and the time spent in the central area in the OFT (Fig. 8c, d). Furthermore, LS-$A_{2A}R$ knockdown also increased the consumption of sucrose compared with the control group (Fig. 8f) without altering the total consumption of liquid (Fig. 8e). Thus, the selective knockdown of $A_{2A}R$ in the LS produced antidepressant-like behaviors.

To explore the translational potential of pharmacological targeting LS-$A_{2A}R$, we determined the ability of the locally applied $A_{2A}R$ antagonist KW6002 (recently approved anti-parkinsonian by US-FDA[28]) to reverse the chronic stress-induced depressive phenotypes (Fig. 9a). Male C57BL/6J mice were first exposed to either CRS or no-stress stimuli for 14 consecutive days. CRS mice with a confirmed depressive-like phenotype (Fig. 9b–d) or control (i.e. non-stressed) mice were randomly assigned for treatment with KW6002 (unilateral intra-LS infusion of KW6002, 0.5 µg/µL, 2 µL per injection) or vehicle (methylcellulose) daily for three consecutive days. As expected, CRS-mice treated with vehicle displayed depressive-like behavior as indicated by increased immobility in the TST. LS infusion of KW6002 not only decreased the immobility time in the non-stress mice, but also reversed the increased immobility time induced by CRS, without affecting either locomotion or the time spent in the center in the OFT (Fig. 9e–g). Importantly, two-way ANOVA analysis indicated that there was a CRS x KW6002 treatment interaction ($p = 0.0436$), indicating that KW6002 preferentially reverted the CRS-induced depressive-like behavior. Thus, the pharmacological blockade of LS-$A_{2A}R$ can reverse the depressive-like phenotype caused by CRS.

## Discussion

The present study allowed identifying the LS-$A_{2A}R^+$ neurons as direct upstream integrative regulators of depressive-like behavior, as supported by four convergent sets of experimental findings: (i) the upregulation of $A_{2A}R$ in the LS in two mouse models of chronic stress; (ii) the sufficiency and necessity of $A_{2A}R$ activation in the LS to produce depressive-like behaviors; (iii) $A_{2A}R$ activation in LS-$A_{2A}R^+$ neurons increased their neuronal firing leading to a suppressed activity of their surrounding LS neurons; (iv) the emergence of depressive-like behavior upon activation of LS-$A_{2A}R^+$ neurons by signaling via their downstream LS-$A_{2A}R^+$ → LHb and LS-$A_{2A}R^+$ → DMH pathways. This cellular and neural circuit dissection prompts a rationale to understand the antidepressant effects of $A_{2A}R$ antagonists, based on a novel working model whereby aberrantly enhanced LS-$A_{2A}R$ signaling mimics stressor signals to trigger depressant behavior: LS-$A_{2A}R^+$ GABAergic neurons sense information from internal and external stressors upregulating $A_{2A}R$ signaling to increase their firing rate and to

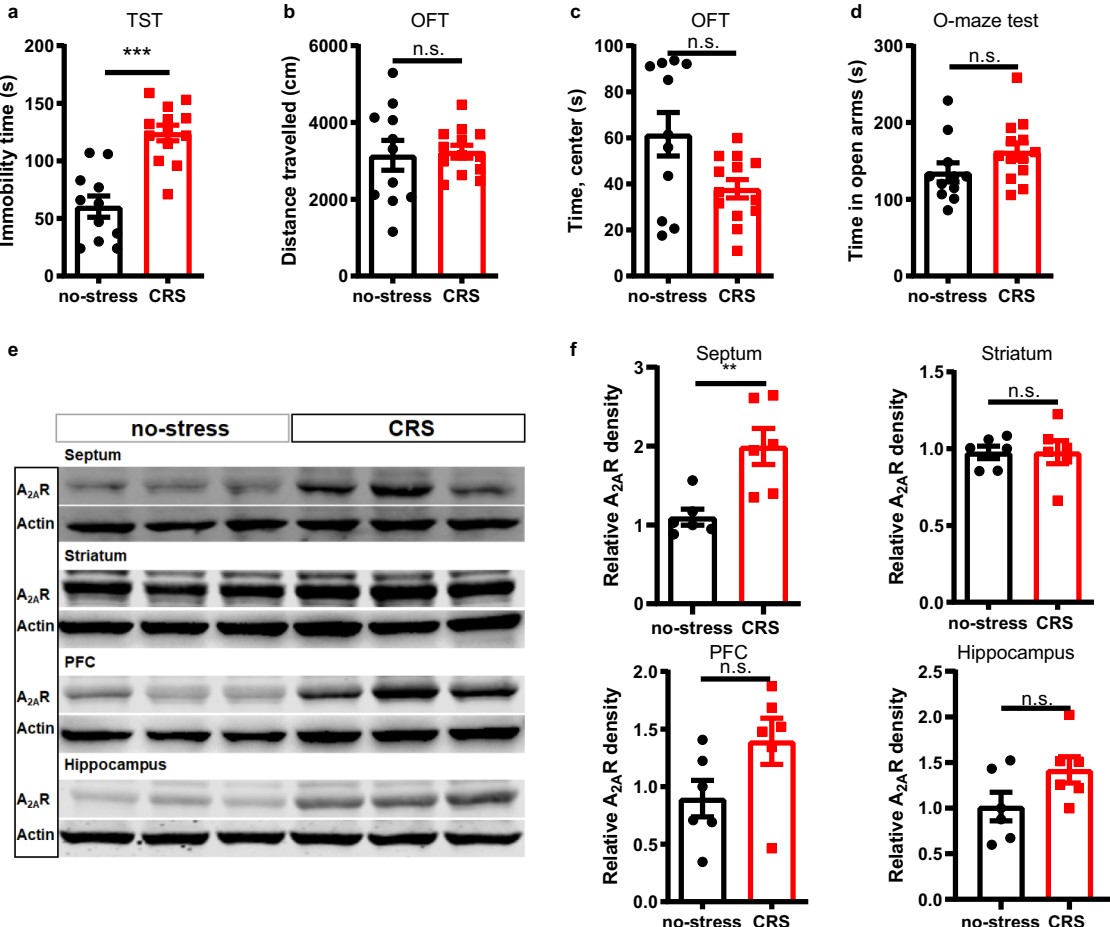

**Fig. 6 | A$_{2A}$R are selectively upregulated in the lateral septum (LS) of mice subject to chronic restraint stress (CRS). a–d** Compared with control mice, CRS mice showed an increase in immobility time in a tail suspension test (TST) ($n = 11$ and 13 mice per group for no-stress and CRS, respectively, Unpaired t test, $p = 0.00001$, t(22) = 5.674) (**a**), without changes in the total distance traveled in an open field test (OFT) ($n = 11$ and 13 mice per group for no-stress and CRS, respectively, Unpaired t test, $p = 0.8169$, t(22)=0.2343) (**b**). CRS mice spent similar time in the center area of the OFT ($n = 11$ and 13 mice per group for no-stress and CRS, respectively, Mann-Whitney test, $p = 0.0821$, U = 41) (**c**) and in the open arms of an elevated O-maze test ($n = 11$ and 13 mice per group for no-stress and CRS,

respectively, Unpaired t test, $p = 0.1165$, t(22) = 1.634) (**d**). **e, f** Representative Western blot and quantification of A$_{2A}$R protein levels in the septum ($n = 6$ mice/group, Unpaired t test, $p = 0.0050$, t(10) = 3.584), striatum ($n = 6$ mice/group, Unpaired t test, $p = 0.9898$, t(10) = 0.01314), prefrontal cortex (PFC) ($n = 6$ mice/group, Unpaired t test, $p = 0.08$, t(10) = 1.948) and hippocampus ($n = 6$ mice/group, Unpaired t test, $p = 0.0886$, t(10) = 1.886) of CRS and control mice. CRS mice displayed a selective upregulation of A$_{2A}$R in the septum without significant changes in the other three brain regions associated with mood processing. Data were shown as mean ± SEM. **$p < 0.01$, ***$p < 0.001$; n.s., no significant difference. Source data are provided as a Source Data file.

simultaneously decrease LS activity and activate the LS-A$_{2A}$R$^+$ → LHb and LS-A$_{2A}$R$^+$ → DMH pathways to trigger depressant-like behaviors.

These findings align with the involvement of LS circuitry in the expression of chronic stress-induced depressive-like behavior. Amongst the complexity of LS circuitry, we teased apart the functional relevance of LS-A$_{2A}$R$^+$ neurons that emerged as key orchestrators of stress-induced depressive-like behavior. LS-A$_{2A}$R$^+$ neurons were essentially GABAergic neurons and the pharmacological activation of A$_{2A}$R increased their firing frequency. This coincides with the ability of A$_{2A}$R to bolster the activity of defined populations of GABAergic neurons in other brain structures such as in the hippocampus[36], central amygdala[37], prefrontal cortex[38], globus pallidus[39], tuberomammillary nucleus[40] or in the nucleus of the solitary tract[41]. The activation of LS-A$_{2A}$R$^+$ GABAergic neurons lead to a suppression of the activity of surrounding LS cells, as concluded by a decreased c-Fos-immunoreactivity. This is in agreement with the general conclusion that the LS acts as a mood regulator exerting a tonic inhibition onto various subcortical nuclei (reviewed in[22,42]), as heralded by the evidence that activation of LS neurons endowed with dopamine D3 receptors rescues early life stress-induced social impairments[43],

activation of Takeda G protein-coupled receptor 5 increases somatostatin-GABAergic neurons of the dorsolateral septum decreasing depressive-like symptoms[26], genetic elimination of neuroligin-2 reduces LS inhibition to stress-induced activation of downstream hypothalamic nuclei reducing avoidance behavior[44] and a 5HT$_{1A}$ receptor agonist increases LS activity to suppress the HPA axis and increase escape behavior in a forced swimming stress model[45].

The observation that the optogenetic stimulation of the LS-A$_{2A}$R$^+$ terminals in the LHb and DMH clearly show that it is the projections of LS-A$_{2A}$R$^+$ neurons to the LHb and DMH that play the key role controlling stress-induced mood alterations. Thus, we now characterized the specific downstream targets of the LS-A$_{2A}$R$^+$ neurons involved in the control of stress-induced mood dysfunction, since it is known that distinct LS neuron populations project to different downstream targets to exert distinct behavioral modulation (reviewed in[22]). Our circuit-level analysis identified the LS-A$_{2A}$R$^+$ → DMH and LS-A$_{2A}$R$^+$ → LHb pathways as the main downstream target for top-down control of depressive-like behavior by LS-A$_{2A}$R. The DMH is a main output of the LS, establishing a feedback loop to control LS circuits through peptidergic signals such as CRF and

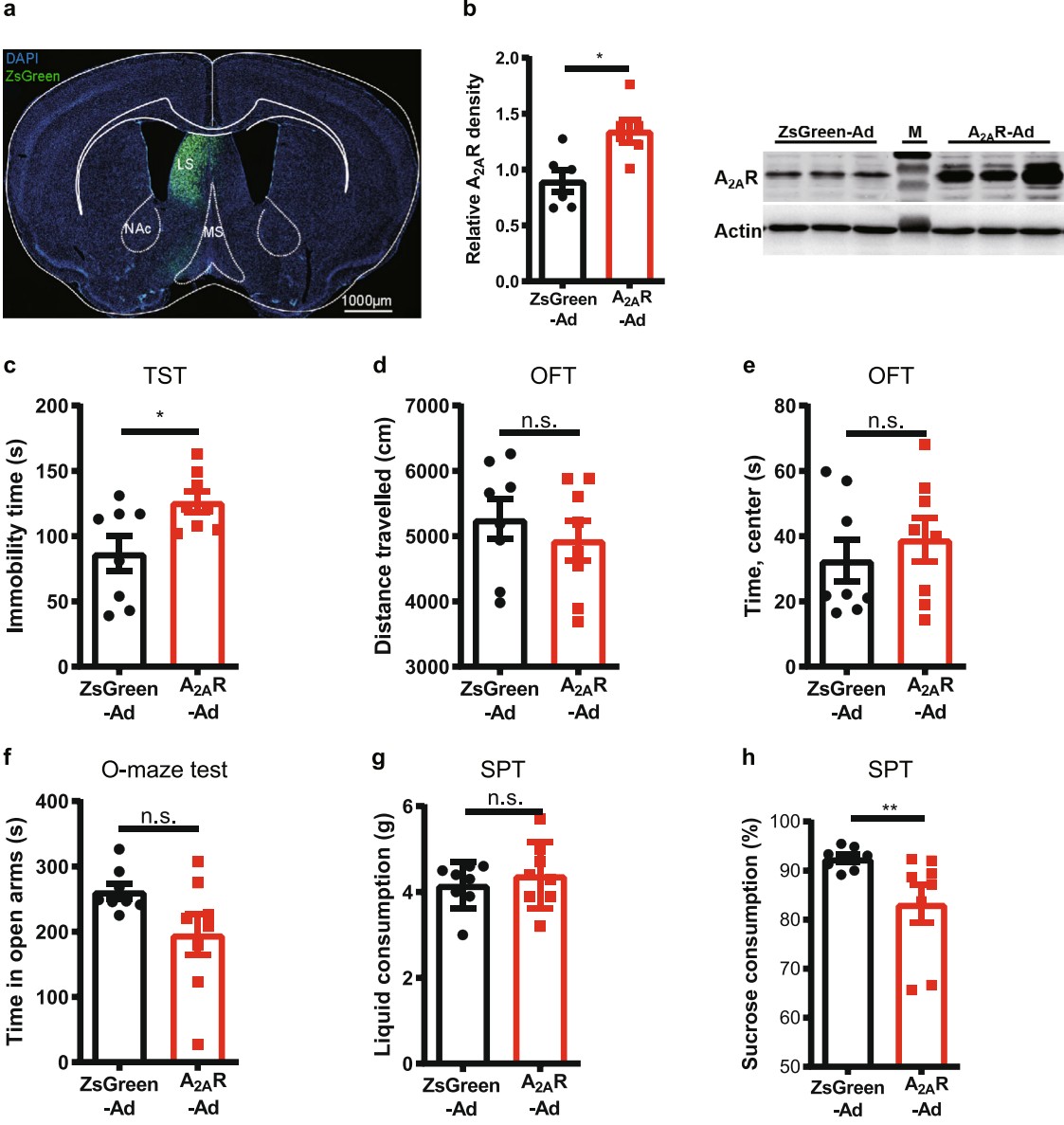

**Fig. 7 | A$_{2A}$R overexpression in the lateral septum (LS) induces depression-like behaviors. a** Adenovirus expressing A$_{2A}$R (A$_{2A}$R-Ad) and ZsGreen (ZsGreen-Ad) in the LS. Nuclei are stained with DAPI in blue. MS, medial septum; NAc, Nucleus Accumbens core. Scale bar: 1000 µm. **b** Representative Western blots and quantification of A$_{2A}$ receptor protein levels of mice injected with A$_{2A}$R-Ad and ZsGreen-Ad ($n = 6$ mice/group, Unpaired t test, $p = 0.0106$, t(10) = 3.135). **c–h** Compared to control mice, A$_{2A}$R-Ad mice displayed increased immobility in the tail suspension test (TST) ($n = 8$ mice/group, Unpaired t test, $p = 0.0229$, t(14) = 2.554) **c**, no change of the total distance traveled ($n = 8$ mice/group, Unpaired t test, $p = 0.4639$,

t(14) = 0.7530) **d** or the time in the central area in the open field test (OFT) ($n = 8$ mice/group, Mann-Whitney test, $p = 0.6454$, U = 27) **e** and in the elevated O-maze test ($n = 8$ mice/group, Unpaired t test with Welch's correction, $p = 0.0784$, Welch-corrected t(8.9) = 1.988) **(f)**. A$_{2A}$R-Ad mice displayed similar total liquid consumption ($n = 8$ mice/group, Mann–Whitney test, $p = 0.7033$, U = 28) **(g)** but a decreased consumption of sucrose compared with the control group ($n = 8$ mice/group, Mann-Whitney test, $p = 0.0070$, U = 7) **(h)** in the sucrose preference test (SPT). Data were shown as mean ± SEM. *$p < 0.05$, **$p < 0.01$; n.s., no significant difference. Source data are provided as a Source Data file.

vasopressin, which is proposed to be involved in depression, fear and anxiety (reviewed in[22,24]). In agreement with this important role of the DMH, our virus tracing using A$_{2A}$R-Cre mice identified the DMH as the main downstream target of LS-A$_{2A}$R$^+$ neurons through a direct projection. Furthermore, activation of LS-A$_{2A}$R$^+$ neurons produced a more robust increase of c-Fos in the DMH. Critically, optogenetic activation of the LS-A$_{2A}$R → DMH pathway recaptured the depressant phenotype of LS-A$_{2A}$R activation. Based on the parallel effects of 5HT$_{1A}$ receptor agonists and of A$_{2A}$R antagonists[18,45] and the proposed heteromerization of both receptors[46], the previously identified bidirectional ability of LS-5HT$_{1A}$ receptors to control the HPA axis and define stress-induced mood alterations[45], prompts the hypothesis that LS-A$_{2A}$R may also control CRH release and HPA axis in the DMH.

This contention provides a tentative mechanism for the recent finding that A$_{2A}$R blockade reverts the depressive-like behavioral, electrophysiological, and morphological alterations induced by early-life maternal separation through a restoration of the activity of the HPA axis[34].

Our circuit-level analysis also identified the involvement of an unreported LS-A$_{2A}$R$^+$ → LHb pathway in the stress-induced expression of depressive-like behavior. This conclusion was based on the combined observations that: (i) virus tracing identified the LHb as a downstream target of LS-A$_{2A}$R$^+$ neurons; (ii) c-Fos expression analysis confirmed the functional activation of LS-A$_{2A}$R$^+$ → LHb pathway upon LS-A$_{2A}$R activation, in accordance with a recently identified stress-induced functional circuit from LS to LHb;[47] (iii) optogenetic activation

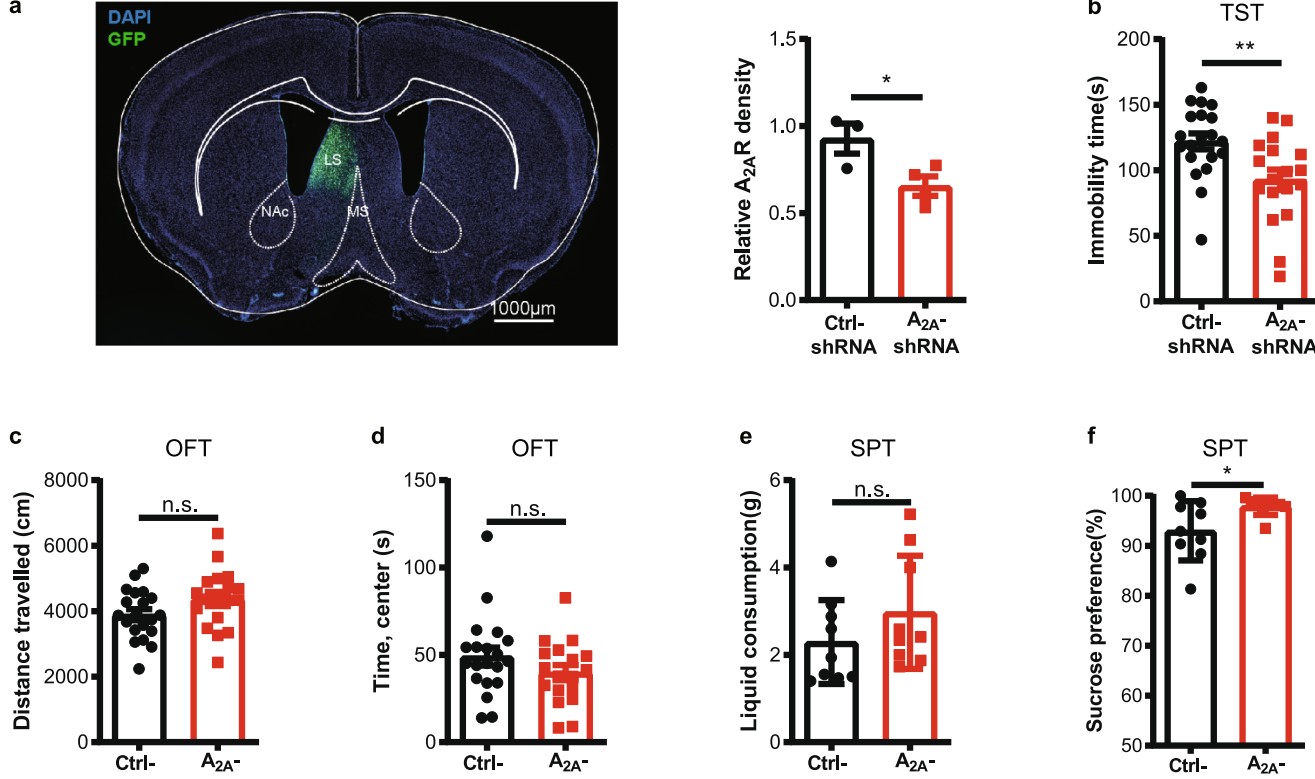

**Fig. 8 | Focal genetic inactivation of A$_{2A}$R in the lateral septum (LS) affords an anti-depressant-like phenotype. a** Left: A$_{2A}$R-specific interference virus (AAV9-syn-A$_{2A}$RshRNA-GFP, A2AR-shRNA) and control virus (AAV9-syn-A$_{2A}$Rshcontrol-GFP, ctrl-shRNA) were unilaterally injected in the LS of C57BL/6 J mice. Nuclei are stained with DAPI in blue. LS, lateral septum; MS, medial septum; NAc, Nucleus Accumbens core. Scale bar: 1000 μm. Right: The downregulation of A$_{2A}$R in the LS was verified by qPCR (*n* = 3 and 4 mice per group for control and A$_{2A}$-shRNA, respectively. Unpaired t test, *p* = 0.0383, t(5)=2.794). **b** The downregulation of A$_{2A}$R reduced immobility in the tail suspension test (TST) (*n* = 20 and 18 mice per group for control and A$_{2A}$-shRNA, respectively, Unpaired t test, *p* = 0.0049, t(36) = 2.995)

without affecting basic motor activity (*n* = 20 and 18 mice per group for control and A$_{2A}$-shRNA, respectively. Unpaired t test, *p* = 0.0860, t(36) = 1.765) and time spent in the central area (*n* = 20 and 18 mice per group for control and A$_{2A}$-shRNA, respectively. Mann–Whitney test, *p* = 0.1645, U = 132) in the open field test (OFT) (**c**, **d**). **e**, **f** Mice subject to LS-A$_{2A}$R knockdown displayed a similar total liquid consumption (*n* = 9 mice/group, Unpaired t test, *p* = 0.2218, t(16) = 1.271) but increased the consumption of sucrose compared with control mice (*n* = 9 mice/group, Mann-Whitney test, *p* = 0.0400, U = 17) in the sucrose preference test (SPT). Data were shown as mean ± SEM. *\*p* < 0.05, \*\**p* < 0.01; n.s., no significant difference. Source data are provided as a Source Data file.

of LS-A$_{2A}$R$^+$ → LHb projections induced depressant effects. The identification of the LS-A$_{2A}$R$^+$ → LHb pathway as a downstream target for LS-A$_{2A}$R control of depressive-like behavior is consistent with the role of the LHb in the development of depression, as heralded by the persistent and robust activity in the LHb of depressed animals[48,49] (reviewed in[50]), by the experimental ability of increased LHb activity to trigger depressive-like behaviors[51–53], by the ability of the fast-acting antidepressant drug ketamine to wane LHb neuronal activity[54] and by the impact of deep-brain stimulation in the LHb to alleviate depressive-like symptoms[55,56]. Although whether the LHb contains GABAergic neurons is still a subject of debate, recent studies[57–59] that locally optogenetic activation of subtypes of GABAergic neurons in the LHb drives inhibitory responses in nearby cells, have indicated the existence of functional local inhibitory circuit within the LHb. Thus, we proposed a disinhibition model whereby LS A$_{2A}$R$^+$ neurons inhibit LHb GABAergic neurons leading to the disinhibition of LHb glutamatergic neurons via a LS$^{GABAergic}$ → LHb$^{GABAergic}$ → LHb$^{glutamatergic}$ circuit. While the specific types of neurons in the LHb controlled by the LS-A$_{2A}$R$^+$ neurons still await to be identified, LS-A$_{2A}$R signaling is concluded to be an upstream regulator of the LHb to implement stress-induced depressive-like maladaptive behavior.

While previous studies have demonstrated that systemic administration of A$_{2A}$R antagonists or genetic inactivation of A$_{2A}$R produces antidepressant effects, the critical site of action of A$_{2A}$R to control depressive-like behavior was unclear. A$_{2A}$R are enriched in the striatum

with a predominant post-synaptic localization to control dopamine signaling[60], with comparatively lower densities in brain regions associated with depression, such as mPFC, hippocampus, and amygdala[61,62], where A$_{2A}$R are mostly presynaptic controlling synaptic plasticity (reviewed in[11]) and synaptic remodeling[63]. Remarkably, extra-striatal A$_{2A}$R have a low density but robustly affects brain function[11], as now also observed for the robust impact of LS-A$_{2A}$R$^+$ neurons in spite of their low abundance. Importantly, altered functioning of neuronal networks leads to a maladaptive up-regulation of A$_{2A}$R, which constitutes a biomarker of progressive synaptic failure and neurodegeneration[12]. This has been documented in hippocampal synapses in models of Alzheimer's disease[64], in cerebrocortical synapses in models of epilepsy[65,66] or of Rasmunsen encephalopathy[67], in corticostriatal synapses in models of early Parkinson's disease[68] or restless leg syndrome[69] or in cerebellar synapses in the model of spinocerebellar ataxia[70] (more information in Supplemental Table 1). We now identified a stress-induced aberrant up-regulation of A$_{2A}$R in the LS (more robust compared with the prefrontal cortex, hippocampus, and striatum) of two mouse models of chronic stress. Overall, these findings prompt LS-A$_{2A}$R as a central adaptive feature relating to stress exposure and A$_{2A}$R-mediated modulation of chronic stress-induced depressive-like behavior.

Considering that the LS receives inputs from various brain regions involved in mood regulation, whether those regions, especially the mPFC, and hippocampus, modulate the activity of the LS A$_{2A}$R$^+$

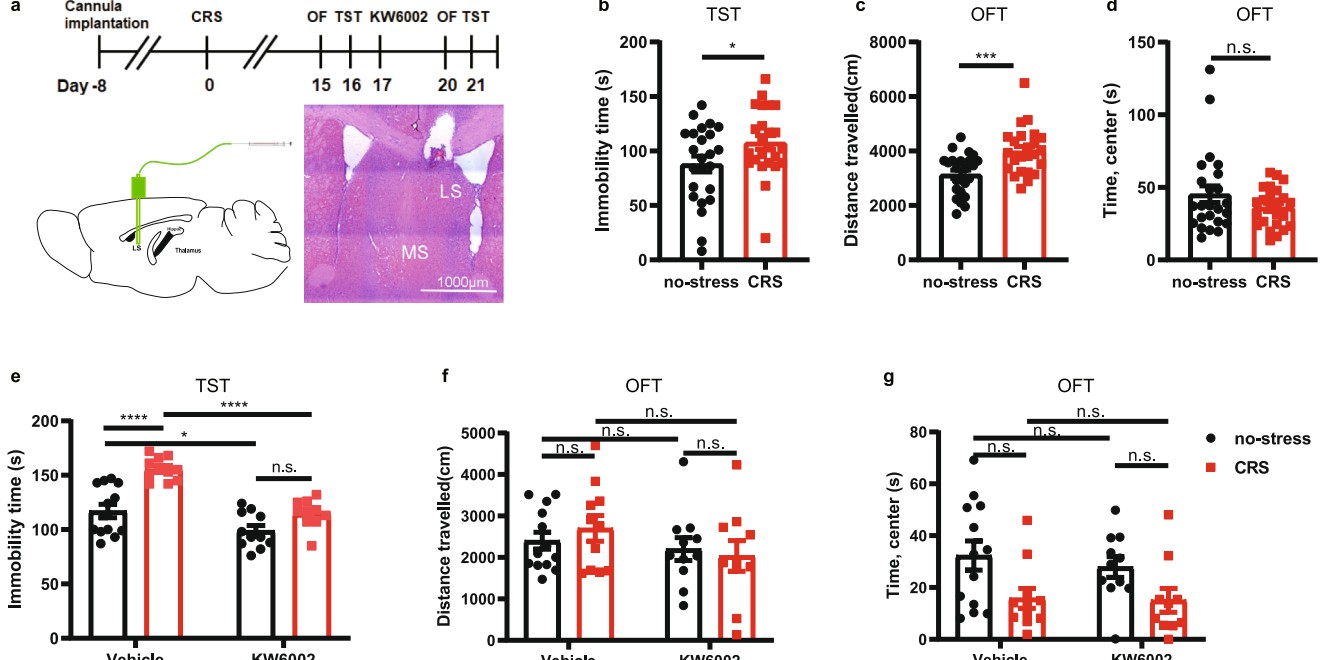

**Fig. 9 | Focal pharmacological inactivation of A$_{2A}$R in the lateral septum (LS) affords an anti-depressant-like phenotype. a** Schematic protocol for investigating the role of the A$_{2A}$R antagonist KW6002 to reverse the CRS-induced depressive phenotypes. Hematoxylin and eosin (HE) staining showing the position of the cannulas. Scale bar: 1000 μm. **b–d** After confirming that the mice subject CRS displayed a depressive-like behaviors in the tail suspension test (TST) ($n = 24$ mice/group, Unpaired t test, $p = 0.0476$, t(46) = 2.035) **b** and in the open field test (OFT) (**c**: $n = 24$ mice/group, Unpaired t test, $p = 0.0006$, t(46) = 3.671; d: $n = 24$ mice/group, Mann–Whitney test, $p = 0.5295$, U = 257), **e–g** injection of KW6002 (0.5 μg/μL, 2 μL) into the LS for three consecutive days reversed the increased immobility in the TST caused by CRS ($n = 13$ (Vehicle: no-stress), 11 (Vehicle: CRS), 11 (KW6002: no-stress) or 10 (KW6002: CRS) mice/group, one-way ANOVA, interaction $p = 0.0269$, F(1,41) = 5.268; $p$(Vehicle: no-stress vs. Vehicle: CRS) = 0.0000071, q(41) = 8.037; $p$(Vehicle: no-stress vs. KW6002: no-stress) = 0.0500, q(41) = 3.787;

$p$(Vehicle: CRS vs. KW6002: CRS) = 0.0000077, q(41) = 8.000; $p$(KW6002: no-stress vs. KW6002: CRS) = 0.1453, q(41) = 3.086) **e**, but did not affect the performance of mice in the OFT ($n = 13, 11, 11$ and 10 samples per group, respectively, one-way ANOVA, interaction $p = 0.4239$, F(1,41)=0.6524; $p$(Vehicle: no-stress vs. Vehicle: CRS) = 0.9736, t(41)=0.7555; $p$(Vehicle: no-stress vs. KW6002: no-stress)=0.9968, t(41) = 0.5051; $p$(Vehicle: CRS vs. KW6002: CRS) = 0.5406, t(41) = 1.581; $p$(KW6002: no-stress vs. KW6002: CRS) = 0.9991, t(41) = 0.3989) (**f**) ($n = 13, 11, 11$ and 10 samples per group, respectively, one-way ANOVA, interaction $p = 0.7041$, F(1,41) = 0.1463; $p$(Vehicle: no-stress vs. Vehicle: CRS) = 0.0674, q(41) = 3.602; $p$(Vehicle: no-stress vs. KW6002: no-stress) = 0.9034, q(41)=0.9643; $p$(Vehicle: CRS vs. KW6002: CRS) = 0.9994, q(41) = 0.1627; $p$(KW6002: no-stress vs. KW6002: CRS) = 0.2594, q(41) = 2.636) (**g**). Data were shown as mean ± SEM. *$p < 0.05$, ***$p < 0.001$, ****$p < 0.0001$; n.s., no significant difference. Source data are provided as a Source Data file.

neurons is still an open question to investigate. The role of the LS in regulating depression has long been established[24], however, the underlying mechanism at the circuit level is largely undefined. The LS receives glutamatergic input predominantly from the hippocampus, which is involved in mood regulation[20]. Recently, Wang et al. have found that silencing and stimulating the circuit from hippocampal CA3 pyramidal neurons to dorsal LS somatostatin-positive, but not PV-positive GABAergic neurons, could bidirectionally regulated depression-like behaviors[26]. To our limited knowledge, available evidence suggests that mPFC outputs to the LS play a key role in modulating depression-like behaviors is lack. Chen et al. have recently found that optogenetic activation of the projection from infralimbic cortex, a sub-region of the mPFC, to the LS promotes anxiety-related behaviors[71], however, whether the mPFC→LS pathway involved in depression needs further investigation.

In summary, we have uncovered and identified aberrantly increased A$_{2A}$R signaling in the LS as a key upstream regulator for stress-induced depressive-like behavior by controlling LS activity and signaling through the LS-A$_{2A}$R$^+$ → DMH and LS-A$_{2A}$R$^+$ → LHb pathways. This understanding of the precise circuit targets of LS-A$_{2A}$R signal as an upstream and integrated regulator of stress-induced depressive-like behavior provides the required rationale to expand the clinical translation of A$_{2A}$R antagonist as a potentially effective anti-depressant, which has already been recently approved by US-FDA for the treatment of Parkinson's disease with a notable safety profile[28].

## Methods

### Study approval

All animal studies were approved by the Institutional Ethics Committee for Animal Use in Research and Education at Wenzhou Medical University, China.

### Animals

Male adult C57BL/6 J mice (8-16 weeks of age, Shanghai JieSiJie Laboratory Animal Co., Ltd.), male or female A$_{2A}$R-tag mice (8-16 weeks of age, Vazyme, Nanjing, China) and A$_{2A}$R-Cre mice (8-16 weeks of age, MMRRC, Stock Number: 031168-UCD) were maintained and used in accordance with protocols approved by the Institutional Ethics Committee for Animal Use in Research and Education at Wenzhou Medical University, China. Mice were housed 3–5 per cage in a suitable temperature (23 ± 1 °C) and relative humidity room (60 ± 2%) under a 12 h light/ dark cycle (light on from 8 a.m. to 8 p.m.) with *ad libitum* food and water, unless otherwise specified.

### Mouse genotyping

Total DNA was isolated from the A$_{2A}$R-Cre or A$_{2A}$R-tag mouse tail using the Dseasy Blood & Tissue Kit (Qiagen, Hilden, Germany). PCR was performed at an annealing temperature of 60 °C using the GoTaq® Flexi DNA Kit (Promega, USA).

The sequences of the primers (synthesized in Sangong Biotech, Shanghai, China) used for A$_{2A}$R-Cre mouse genotyping were

(A$_{2A}$R-Cre forward) 5'-cgtgagaaagcctttgggaagct-3' and
(A$_{2A}$R-Cre reverse) 5'- ccccagaaatgccagattacgtt-3'.

The sequences of the primers used for A$_{2A}$R-tag mouse genotyping were
(A$_{2A}$R-tag forward) 5'-agaccttccggaagatcatccga-3' and
(A$_{2A}$R-tag reverse) 5'- tggggagagtagtgtattagcagg-3'.

### Viral vectors

The following viral constructs were used: AAV2/9-hSyn-A$_{2A}$R-3fxflag-ZsGreen (titer: 1.5E + 12 vector genome (v.g.)/mL, Hanbio Biotechnology, Shanghai, China), AAV2/9-hSyn-ZsGreen (titer: 1.8E + 12 v.g./mL, Hanbio Biotechnology), AAV2/9-A$_{2A}$RshRNA-GFP (titer: 1.5E + 12 v.g./mL, Taitol Biotechnology, Shanghai, China), AAV2/9-A$_{2A}$Rshcontrol-GFP (titer: 1.5E + 12 v.g./mL, Taitol Biotechnology), rAAV2/9-Ef1α-DIO-ChR$_2$(H134R)-EYFP (titer: 2.0E + 12 v.g./mL, BrainVTA, Wuhan, China), rAAV2/9-Ef1α-DIO-eNpHR3.0-EYFP(titer: 5.94E + 12 v.g./mL, BrainVTA), rAAV2/9-Ef1α-DIO-EYFP (titer: 2.0E + 12 v.g./mL, BrainVTA), rAAV2/9-hSyn-DIO-EYFP (titer: 5.14E + 12 v.g./mL, BrainVTA).

### Stereotaxic surgery, AAVs microinjection and optical fiber implantation

For stereotaxic surgery, animals were anesthetized with Avertin (250 mg/kg, Sigma-Aldrich). Viruses were ipsilateral or bilaterally injected by a pressure microinjector with a pulled glass capillary into the different brain regions according to the different experiments, namely: LS ( + 0.8 mm AP; ± 0.35 mm ML; −2.75 mm DV from the brain surface), LHb (−1.72 mm AP; ± 0.46 mm ML; −2.7 mm DV), DMH (−1.70 mm AP; ± 0.5 mm ML; −5.0 mm DV). Volumes of virus ranged between 100-200 nL, administered at the rate of 20 nL/min per hemisphere. The capillary was left in place for 5-10 min after the injection. Mice were put on a heat pad to recover from anesthesia after the surgery. The viral injection sites were verified by post-immunohistochemistry analysis.

For optic fiber implantation, 200 μm diameter optic fibers (Newdoon Technology, Hangzhou, China; or Inper, Hangzhou, China) were implanted 0.05 mm above the virus injection site and fixed to the skull using dental cement.

### Cannula infusion experiment

The cannula (customized by Kedou Brain-computer Technology Co., Ltd, Suzhou, China), constituted by a hollow catheter (0.5 mm outer diameter) and a protective cap containing the inner core (outer diameter of 0.25 mm), was ipsilateral implanted into the LS of male C57BL/6 J mice (8 weeks of age). After the surgery, a double dummy cannula with a 0.5 mm extension beyond the end of the guide cannula with a metal cap, was inserted into the guide cannula. One week after surgery, CGS21680 (0.5 μg/μL) or vehicle (DMSO + saline) was microinjected (at the rate of 1 μL/min) into the LS of freely moving mice with a drug delivery inner core connected to the microinjection pump. For KW6002 treatment, KW6002 (istradefylline; 0.5 μg/μL) dissolved in 0.5% methylcellulose, synthesized as described previously[72]) or vehicle (methylcellulose + saline) was microinjected (at a rate of 1 μL/min) into the LS of freely moving mice for 3 consecutive days after the 14 days of CRS. The drug (2 μL) was then infused into the LS though the cannula. The injector cannula inner core was left in the LS for an additional 5 min to allow adequate local drug diffusion and minimize the spread of the drug along the injection track. Then mice were exposed to behavioral tests. Only data from mice with a correct site of injection, confirmed by hematoxylin/eosin (HE) staining (C0105S, Beyotime Biotech Inc, Shanghai, China), were used.

### Optogenetic manipulations

For A$_{2A}$R-Cre mice expressing ChR2 or EYFP, a 465 nm blue light laser (Newdoon Technology) was used. Light intensity was calculated to be

4−5 mW, which was estimated at 4 mm from optic patch cable tip with a ceramic sleeve. Blue light was delivered at 20 Hz with 20-ms pulses. For A$_{2A}$R-Cre mice expressing eNpHR3.0 or EYFP, a laser delivered 589-nm yellow light (Newdoon Technology) at 0.0001 Hz, with 9999 s pulses. Light stimulation was carried out during the whole duration of the behavioral test.

### Animals stress models

Male C57BL/6 J mice (8 weeks of age) were used in the stress-related experiments. Before starting any stress paradigm, mice with matched age and weight were randomly assigned into the non-stress or stress groups.

**Chronic restraint stress (CRS).** CRS was performed in 50 mL centrifuge tubes with holes for ventilation. Mice were restrained horizontally in tubes for 3 h in the first 7 days and for 4−5 h in the next 7 days (up to 14 days)[54]. Non-stressed mice were left undisturbed in their home cages.

**5 days Repeated Forced Swim Stress (5d-RFSS).** Mice were subjected to repeat swimming in a transparent cylinder (15 cm diameter, 25 cm height) containing 20 cm of water (22−25°C) for 10 min daily for 5 consecutive days (induction phase). From day 6 on, the mice were kept in their home cage without swimming for 4 weeks, after which a last swim was imposed on day 32 (test phase)[35].

### Behavioral tests

All behavior tests were performed between 10:00 AM and 5:00 PM in a sound attenuated room and were recorded on videotape for offline analysis with the EthoVision XT system. Between trials, the room and apparatus were cleaned with 70% ethanol. All behavioral experiments were carried out with the experimenter blind to genotype and/or treatment history.

**Open field test (OFT).** Animals were individually placed in the center of a chamber (40 ×40 x 40 cm) in a soundproof environment with gentle light and their movement was analyzed during 10 min.

**Tail suspension test (TST).** The mice tails were wrapped with tape at approximately 1 cm from the end of the tail. The mice were then fixed upside down on a horizontal bar with the nose tip about 30 cm above the ground. Animal behaviors were recorded for 6 min and the immobility time was scored during the last 4 min.

**Elevated O-maze.** The maze was constructed in a circular track 10 cm wide, 105 cm in diameter, and elevated 72 cm from the floor. The maze was divided in four quadrants of equal length with two opposing open quadrants with 1 cm high curbs to prevent falls and two opposing closed quadrants with walls 28 cm in height. A 10 min trial under gentle light conditions was carried out with the animal placed in the center of a closed quadrant to analyze their movement.

**Forced swim test (FST).** Animals were individually placed in vertical clear glass cylinder (20 cm in diameter, 30 cm in height) filled with water (21−25°C). Water depth was set to prevent mice from touching the glass bottom with their limbs or tails. The test lasted for 6 min and the immobility time was counted from 2 to 6 min. Mice were regarded as immobile when floating motionless or making only movements that were necessary to hold its head above the water.

**Sucrose preference test (SPT).** Mice were single housed and habituated with 1% sucrose and water for 2 days and the bottle positions were counterbalanced every 12 h. On the testing day, mice were water and food-deprived for 12 h and then exposed to pre-weighed identical bottles (one bottle of water and one bottle of 1% sucrose) for 12 h in the

dark phase. Sucrose preference was calculated by dividing the consumption of sucrose by the total liquid consumption (water and sucrose).

## Western blots

For Western blot analysis, brains were quickly removed from euthanized C57BL/6 J mice and different brain regions (septum, striatum, PFC and hippocampus) were carefully dissected. Tissues were lysed by sonication in ice-cold RIPA lysis buffer (Beyotime) with complete a protease inhibitor cocktail (Beyotime) and phosphatase inhibitors mix (Bimake, Houston, USA), incubated on a roller for 30 min at 4 ˚C and cleared by centrifugation at 17530 g for 15 min. The supernatant was collected and the protein concentration was estimated using Enhanced BCA Protein Assay Kit (Beyotime). Samples were diluted in 5x SDS sample buffer and analyzed by SDS-PAGE. Mouse monoclonal anti-$A_{2A}R$ (1:200, 7F6-G5-A2, Santa Cruz Biotechnology) or rabbit anti-actin (1:3000, 66009-1-Ig, Proteintech) antibodies were used to evaluate the relative amount of $A_{2A}R$.

## Immunofluorescence staining

For immunofluorescence staining, 60 min after injecting CGS21680 into the LS, mice were anesthetized and perfused with PBS followed by 4% paraformaldehyde (PFA) in PBS (pH 7.4). The brains were quickly removed, post-fixed in 4% PFA overnight, and then dehydrated in 30% sucrose solutions in PBS for 3 days. The brain was sectioned to a thickness of 30 μm using a cryostat (Leica CM1950) and preserved in PBS. Free-floating sections were blocked in blocking solution (0.3% Triton X-100 in PBS and 5% normal donkey serum) for 1 h at room temperature. Sections were then incubated with the primary antibody in antibody solution (5% normal goat serum, 0.3% Triton X-100, 1% bovine serum albumin in PBS) overnight at 4 °C. Sections were then washed with PBS (3 × 10 min) and incubated for 2 h at room temperature with the secondary antibodies and DAPI. Finally, sections were washed with PBS (3 × 10 min). According to the anatomical location of each brain area (identified with the 4th edition of *The mouse brain in stereotaxic coordinates*), nine sections containing LS, LHb, and DMH were selected for counting. The following antibodies were used: rabbit anti-c-Fos (1:1000, PC05, EMD Millipore); donkey anti-rabbit 488 (1:500, A-21206, Invitrogen). The anti-HA antibody (1:200, A-11003, Invitrogen) was used to label $A_{2A}R$ in $A_{2A}R$-tag mice. The acquisition of fluorescent images was performed with a Leica DM6B microscope or a Zeiss LSM 880 NLO confocal microscope. For immunohistochemistry quantification, sample IDs were renamed to render the experimenter 'blind' to the analysis.

For immunofluorescence staining after in vitro electrophysiology recording, 300 μm slices were collected and post-fixed in 4% PFA overnight. Free-floating sections were blocked in blocking solution (0.3% Triton X-100 in PBS and 10% normal donkey serum) for 1 h at room temperature and then incubated with rabbit anti-GAD65 + 67 antibody (1:200, ab183999, Abcam) in antibody solution (0.3% Triton X-100 in PBS and 5% normal donkey serum) overnight at 4 °C. After washing with PBS, the sections were incubated with secondary antibodies (donkey anti-rabbit 488, 1:500, A-21206, Invitrogen, or Texas Red Avidin D, 1:500, A-2006-5, Vector Laboratories).

## RNAscope in situ hybridization

RNAscope (Advanced Cell Diagnostics, Hayward, California) was used for ultrasensitive detection and visualization of weakly expressed mRNAs in the LS. All mouse-specific probes for $A_{2A}R$, Slc32a1, and 3-plex negative control probe were synthesized by the manufacturer. Mice were deeply anesthetized with Avertin before quickly removing their brains. After fixing with 4% PFA, brains were cut with a freezing microtome (Leica) into 12 μm sections, adhered to SuperFrost Plus slides (Epredia), and immediately refrozen at −80 °C. Positive (mouse Ppib, Peptidylprolyl Isomerase B) and negative (DapB, 4-hydroxy-

tetrahydrodipicolinate reductase) control probes were included in each experiment. The samples were processed according to RNA-scope® Multiplex Fluorescent Assay manual. Briefly, sections underwent two steps of pretreatment, including a 10-min step of protease digestion. Hybridization with specific probes was then performed for 2 h at 40 °C, followed by three steps of amplification. Opal fluorescent 520 and 570 were used to detect the chromogen in the exposure step. Two washes of 2 min were observed between each amplification step. Image acquisition was performed with Zeiss LSM 880 NLO confocal microscope.

## RNA extraction and quantitative real-time PCR

Total RNAs was extracted according to the manual, using Trizol reagent (Invitrogen). Reverse transcription was carried out using PrimeScript™ RT Master Mix (Takara, Japan). The reaction mixture was incubated for 10 min at 25 °C, followed by 50 min at 37 °C, then heat inactivated at 70 °C for 15 min. Real-time PCR was performed with StepOne Real-time PCR system by iTaq Universal SYBR Green Supermix (Bio-RAD). Relative LS $A_{2A}R$ expression levels were calculated by the comparative CT method. Q-PCR primers (synthesized in Sangong Biotech) used were:

A$_{2A}$R Forward primer: 5′-CCGAATTCCACTCCGGTACA-3′
A$_{2A}$R Reverse primer: 5′-CAGTTGTTCCAGCCCAGCAT-3′

## Acute slice preparation

rAAV2/9-hSyn-DIO-EGFP virus was injected into LS of $A_{2A}R$-Cre mice (aged 8 weeks). Two to three weeks later, mice were anesthetized with 3% isoflurane, perfused with oxygenated high sucrose and ice-cold slicing solution (in mM: 234 sucrose, 11 glucose, 26 NaHCO$_3$, 2.5 KCl, 1.25 NaH$_2$PO$_4$, 10 MgSO$_4$, and 0.5 CaCl$_2$; gassed with 95% O$_2$/5% CO$_2$). The brains were quickly removed and placed into the oxygenated and ice-cold high sucrose slicing solution for 2-3 min. Coronal slices (300 μm thick) containing the LS (AP + 0.8 mm) were sectioned in ice-cold slicing solution with a Leica VT1200S vibratome, then incubated in aCSF (in mM: 126 NaCl, 26 NaHCO$_3$, 10 glucose, 2.5 KCl, 1.25 NaH$_2$PO$_4$, 2 MgCl$_2$, and 2 CaCl$_2$; pH 7.4, bubbled with 95% O$_2$/5% CO$_2$) 33 °C for 1 hour. Before recordings, slices were allowed to recover for at least 1 h in aCSF continuously gassed with 95% O$_2$/5% CO$_2$.

## In vitro electrophysiology

Patch clamp recordings were made from LS neurons under visualization using an upright microscope (Olympus BX51XI) equipped with a 5x objective and a 40x water-immersion objective, infrared (IR) illumination, Nomarski optics, and an IR-sensitive video camera (Electro, Canada). $A_{2A}R^+$ neurons with green fluorescence were visualized using a mercury lamp (Olympus, U-RFL-T) through a GFP filter on the microscope. Light or fluorescent imaging data were acquired using OCULAR software (version 2.0). During recording, coronal slices containing LS area were continuously perfused at 2-3 mL/min with oxygenated aCSF and maintained at 32 °C. The patch pipettes were pulled with a pipette puller (PC-100, Narishige) from borosilicate glass (Sutter Instrument) and had a resistance of 5-6 MΩ. Electrophysiological recordings were made using a Multiclamp 700B amplifier. Signals were amplified, filtered at 2 kHz, and sampled at 10 kHz using Digidata 1550B. Clampfit 10.6 (Molecular Devices) was used to analyze offline electrophysiological data, and Mini Analysis Program 6.0.7 (Synaptosoft Inc., New Jersey) was used to confirm the cell-attached action potential firing rate after Clampfit analysis, with the filtering value set to 100pA for action potential amplitude analysis.

A cell-attached mode was used to assess the effects of $A_{2A}R$ ligands on the excitability of LS-$A_{2A}R^+$ neurons. Glass pipettes were filled with intracellular solution (in mM: 130 K-gluconate, 10 KCl, 10

HEPES, 10 EGTA, 2 $MgCl_2$, 2 $Na_2ATP$, 0.3 $Na_3GTP$). While approaching the cell, positive pressure was applied to the patch pipette. The seal (10 MΩ – > 5 GΩ) between the recording pipette and the cell membrane was obtained by applying suction to the electrode. Action potentials were recorded in cell-attached mode, with the holding potential set as 0 mV. Resting membrane potential and cell action potential firing properties were recorded in current-clamp mode, with cells were maintained at 0 pA holding current. The IPSCs were recorded in voltage clamp mode ($V_{Hold}$ = −60 mV) with continuous bath application of aCSF and glutamate receptor blockers 6,7-dinitroquinoxaline-2,3-dione (DNQX, 20 μM, Tocris) and DL-2-amino-5-phosphonopentanoic acid (DL-AP$_5$, 100 μM, Tocris). CGS21680 (30 nM) was added through bath application with DNQX and DL-AP$_5$ in aCSF.

In order to investigate the morphology of each recorded cell, biocytin (1%, B1592, ThermoFisher Scientific) was added to the intracellular solution, and cells were kept under whole-cell recording mode for at least 10 minutes, allowing the biocytin to diffuse into the patched cell.

### Single-cell reverse transcription quantitative PCR (RT-qPCR)

The cytoplasm of recorded cells was harvested by glass pipettes and expelled into 0.5 mL Eppendorf tube containing 2 μL of 5x PrimeScript RT Master Mix (Takara) and 6 μL RNase Free dH$_2$O. Intracellular pipette solution and RNase Free dH$_2$O were used as controls during the reverse transcription. This mixture was incubated for 10 min at 25 °C, followed by 50 min at 37 °C, then heat inactivated at 70 °C for 15 min, using the entire cDNA template reaction (10 μL) in the first round of PCR amplification. Each of the first round of reaction was prepared as follows: GAPDH F1 1 μL, P1 1 μl; VGAT F1 1 μL, P1 1 μL; cDNA 10 μL, mix 10 μL, ddH$_2$O 26 μL. For the second round of PCR, the nested primers (internal to the first-round pairs) were used. Each of the second round of reaction was prepared as follows: GAPDH F1 1 μL, P2 1 μL; VGAT F1 1 μL, P2 1 μL; cDNA 1 μL, mix 10 μL, ddH$_2$O 5 μL. The thermal cycling program of the second round was: 94 °C for 1 min, 30 cycles of amplification representing three steps of denaturing, annealing and extension (94 °C for 30 s; 55 °C for 1 min; 72 °C for 1 min) and a final extension at 72 °C for 7 min. Amplification products were analyzed on a 2% agarose gel and visualized by staining with ethidium bromide.

RT-PCR primers (synthesized in Sangong Biotech) used were:
GAPDH Forward primer: TGGAAAGCTGTGGCGTGAT
GAPDH Reverse primer1: GTTGCTGTTGAAGTCGCAGG 292
GAPDH Reverse primer2: GACGGACACATTGGGGGTAG 151
GAD67 Forward primer: AATACTACCAACCTGCCGCCC
GAD67 Reverse primer1: CCCGTTCTTAGCTGGAAGCA 270
GAD67 Reverse primer2: AACAGGTTGGAGAAGTCGGTC 250

### Statistics and reproducibility

In all behavioral, electrophysiological and molecular experiments, mice were randomly grouped and the offline data statistical analysis was performed blindly using SPSS 16.0 or GraphPad Prism 8.0 (GraphPad, San Diego, California). If viral infection or drug delivery was not correctly sited, the data points were excluded from analyzes. Animal or replicate numbers for each experiment and results of the statistical analyzes, including exact p-values and degrees of freedom are mentioned in the Figure legends. Experiments for western blot and brain morphological analysis were repeated independently with similar results for at least three times. All data were checked for normality and homogeneity of variance. Student's t-test (unpaired or paired), Mann-Whitney test, or Wilcoxon test was used to compare means between two groups, and two-way analysis of variance followed by Tukey's or Bonferroni *post hoc* tests were used to determine significant differences among multiple groups. All tests were two-sided, with a confidence level of 95%. Differences were considered significant if $p < 0.05$. All data were presented as mean ± SEM. No statistical methods were used to predetermine sample sizes, but our sample sizes are similar to those generally employed in comparable studies.

### Reporting summary

Further information on research design is available in the Nature Portfolio Reporting Summary linked to this article.

## Data availability

The data supporting the findings of this study are included in the figures and supporting files. The raw data are available from the corresponding author upon request. Source data are provided with this paper Source data are provided with this paper.

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

## Acknowledgements

We thank Dr. Jian-jun Zhang (Institute of Psychology of the Chinese Academy of Sciences) and Dr. Chun Hu (South China Normal University) for thoughtful comments for the manuscript. This work was supported by the National Natural Science Foundation of China (Grant No. 31970948 and No. 31600859 to W.G.; No. 81871035 and No.82071378 to P.L.; No. 82101556 to X.Z.), the Research Fund for International Senior Scientists (Grant No. 82150710558 to J.C.), Zhejiang Provincial Natural Science Foundation (Grant No. LZ19H090001 to P.L.; No.LQ22H090013 to X.Z.), the Project of State Key Laboratory of Ophthalmology, Optometry and Vision Science, Wenzhou Medical University (J01-20190101 to J.C.), La Caixa Foundation (LCF/PR/HP17/52190001 to R.C.), Centro 2020 (CENTRO-01-0246-FEDER-000010 to R.C.) and FCT (POCI-01-0145-FEDER-03127, UIDB/04539/2020 and IF/01492/2015 to R.C.). And thanks to Mr. Jiefeng Zhu for helping to draw the beautiful painting.

## Author contributions

Conceptualization: W.G., M.W., and P.L.; Methodology: W.G., M.W., P.L., and ZW.L.; Formal analysis: W.G., M.W., B.S., ZW.L., W.Z., Y.H., T.X., and R.C.; Investigation: W.G., M.W., B.S., ZW.L., W.Z., Z.X., Y.H., T.X., C.C., L.D.; Resources: Y.D., M.Y., ZQ.L., J.Z., X.Z., F.Y.; Writing - Original Draft: W.G., M.W., P.L.; Writing - Review & Editing: W.G., J.C., R.C.; Visualization: M.W., B.S., ZW.L.; Supervision: J.C.; Project administration: W.G., J.C.; Funding acquisition: W.G., J.C., P.L., R.C., X.Z.

## Competing interests

The authors declare no competing interests.

## Additional information

[1]The Molecular Neuropharmacology Laboratory and the Eye-Brain Research Center, The State Key Laboratory of Ophthalmology, Optometry and Vision Science, Wenzhou Medical University, Wenzhou, China. [2]Department of Neurology, The Second Affiliated Hospital and Yuying Children's Hospital of Wenzhou Medical University, Wenzhou, China. [3]Key Laboratory of Structural Malformations in Children of Zhejiang Province, Wenzhou 325000 Zhejiang Province, China. [4]Faculty of Medicine, University of Coimbra, 3004-504 Coimbra, Portugal. [5]Portuguese National Institute of Legal Medicine and Forensic Sciences (INMLCF, IP), Coimbra, Portugal. [6]Department of Neurobiology, Key Laboratory of Molecular Neurobiology, Ministry of Education, Naval Medical University, Shanghai, China. [7]CNC-Center for Neuroscience and Cell Biology, University of Coimbra, 3004-504 Coimbra, Portugal. [8]Oujiang Laboratory (Zhejiang Laboratory for Regenerative Medicine, Vision and Brain Health), School of Ophthalmology & Optometry and Eye Hospital, Wenzhou Medical University, Wenzhou, China. [9]These authors contributed equally: Muran Wang, Peijun Li, Zewen Li. ✉e-mail: chenjf555@gmail.com; guoweihaha@126.com

