## [Peer Review File · Nature Communications]

Lateral septum adenosine A2A receptors control stress-induced depressive-like behaviors via signaling to the hypothalamus and habenulaREVIEWER COMMENTS

Reviewer #1 (Remarks to the Author):

The paper by M Wang and co-workers is entitled “Lateral septum adenosine A2A receptors control stress-induced depressive-like behaviors via signaling to the hypothalamus and habenula”.

The results obtained in this paper are extremely interesting and convincing.

The authors show that a subpopulation of lateral septum (LS) GABAergic adenosine A2A receptors-positive neurons mediate depression-like effects in mice via projections to the dorsomedial hypothalamus (DMH) and to the lateral habenula (LHb).

They show that the optogenetic activation of LS adenosine A2A receptor-positive neurons projection terminals to the DMH or to the LHb induced depressive-like behaviors in the tail suspension test (TST), without inducing locomotor effects or anxiety-like effects.

It is noteworthy that a selective upregulation of A2A receptors in the LS was observed in two models of stress in mice and was also observed in postmortem human brains of suicide completers.

Finally, local injection in the LS of the selective A2A receptor antagonist KW6002 and knockdown of A2A receptors reduced immobility in the TST without affecting motor activity.

There are however some points that might be considered to improve the manuscript.

Introduction:

Classically, it is acknowledged that ventral tegmental area (VTA), medial prefrontal cortex (mPFC), lateral habenula (LHb), lateral septum (LS), amygdala and hippocampus are implicated in major depressive disorders as quoted by the authors (lines 111-113). Therefore, the reader would probably appreciate to read in the discussion section information about the connections of LS GABAergic adenosine A2A receptors-positive neurons with these brain regions, especially mPFC and hippocampus. See PMID: 35443088 DOI: 10.1002/jnr.25052

Does CGS21680 infusion into the LS induce an upregulation of c-Fos expression in these structures?

Results: Figure 5. Optogenetic activation of the projection terminals in the dorsomedial hypothalamus (DMN) induced a more robust increase in immobility in tail suspension test (TST) than in lateral habenula (LHb). This could be discussed more thoroughly.

Discussion:

The classical view is that metabotropic Glu2/3 receptor antagonists display antidepressant-like effects, not agonists. So how can this fit with the sentence line 306-307?

Same paragraph: 5-HT1A agonists are potential antidepressants so the citation lines 311-312 and 326-

328 is very interesting but the reference quoted is probably wrong.

The hypothesis of a restoration of the activity of the HPA axis is indeed interesting and fits well with the data of Batalha (2013-2016). It is unknown whether local knockdown of A2AR or KW6002 infused in LS impact corticosterone levels following stress and status of GC receptors.

In my opinion the sentences 357-361 are not very informative in the context of the study and might be substituted by a figure (see for example those drawn in the review by Patel. PMID: 35443088 DOI: 10.1002/jnr.25052).

Finally, in the current study only male mice were used. It is known that stress responses vary across sex and may underlie the heightened vulnerability to psychopathology in females. See:

PMID: 29316846 DOI: 10.1080/10253890.2017.1422488

PMID: 34404738 DOI: 10.1101/cshperspect.a039198

PMID: 28825715 DOI: 10.1038/nm.4386

PMID: 29548746 DOI: 10.1016/j.biopsych.2018.01.017

So, authors could investigate whether optogenetic suppression of LS A2AR-positive neurons decrease the immobility time in the TST also in females.

Reviewer #2 (Remarks to the Author):

The main idea of this manuscript of the circuit dissection is very good and the selected approach is very elegant. However, many important methodological details were overlooked (namely regarding photo-stimulation) and the results are over-interpreted as a consequence of some crucial controls being absent.

Major issues:

Methods: There is no explanation whatsoever of the photo-stimulation methods, which is key to validate the results from optogenetic manipulations. Without this information (light power, stimulation frequency, etc.) is not possible to evaluate these data properly.

of cells that are LS-A2R? is it a biological relevant enough number?

Appropriate description of controls in each of the experiments is missing, especially since all this is extremely minute.

If LS-A2A+ neurons have their projections at the Lhb and DMH, why are the authors they measuring local effects in figure 2F?

CGS infused into LS will change cfos expression in target neurons in two ways:

1. direct A2A activation in LS-A2A+ neurons OR
 2. activation of A2AR in afferent pathways of the LS, that then also affect their target regions....
- The infusion experiment does not allow any selective discrimination of the source of cell-mediate effects

The western blot is used to claim a selective LS overexpression of A2A in the stress model, however that is not supported by the figure

Please find below the detailed revision:

ABSTRACT:

Line 48 – the sentence of ‘depression being the single largest contributor...’ is not clear or objective. Should be something like ‘Depressive disorders rank as the #1 cause of disease burden worldwide’

Line 49 – the claim of high non-responsiveness to anti-depressive treatments is not correct. Must be rephrased.

Line 52 - Neurons and especially receptors do NOT mediate depression which is a disease with various symptoms...

Line 55 – It is not clear what: ‘formatted depressive phenotype’ means

Line 57 – Error: should be we have shown and NOT we shown

Line 59 – What is the meaning of ‘suicide completers’ ?

Line 117 – Please revise the sentence: ‘mediating depression via direct projects to...’

Line 126 – There is mention to a previous study but the reference is missing.

RESULTS:

Line 150 - These are non-labeled neurons presumably also Gabaregic? Or not?

Fig1A. The density of neurons seems too low, the quality of images is not good enough to see terminals, especially in LHb. A quantification of both, infected cells and terminals throughout different brain areas not restricted to the ones they show, could strengthen the data and give a real idea of the relative importance of these areas as LS output.

Fig 1B. The quality of images should be improved, and quantifications could also help. There are areas of fluorescence outside the LH and DMH, for example in the dentate gyrus. This looks like the tract of the injection, but how could GFP be expressed in areas not targeted by the cre-aav infected in LS? If LS cre-expressing neurons target DG, it should be considered as a possible downstream effector when doing optogenetic experiments.

Fig2A. Not clear which image areas are magnified in each magnification. What is the massive fluorescent label laterally to the ventricle, and outside the LS?

Fig2B. Very hard to see GAD label, improving images quality could help. Also, this is a single examples. It would be helpful to show immunos from slices with several GFP labeled neurons expressing GAD, and a quantification across slices/mice.

Fig2C. Not clear if this is from a single cell or pooled from many cells, which could be confusing.

Figure 2F – The authors claim that ‘We found that A2AR activation by CGS21680 (30 nM) reduced the frequency (but not amplitude) of spontaneous inhibitory postsynaptic currents (sIPSCs) of neurons around EYFP-positive cells, which likely resulted from the increase of A2AR+ GABAergic inputs to non-A2AR positive neurons in the LS (Figure 2F).

This claim is not clear, why would the frequency of inputs to A2AR- neurons decrease when neighboring GABAergic A2AR+ neurons are highly increasing their firing rate? If these neurons are connected to each other, as suggested by the authors, the frequency of sIPSCs should increase. This provided that A2AR+ neurons actually synapse onto A2AR- neurons which is not shown. Pair-recordings or optogenetic experiments are needed to show that this connection is real.

If you are postulating that A2A+ neurons project onto LHb and DMH, then how do you reconcile this with the local circuit effects they are claiming here to be due to the neurons that express A2A receptors onto neurons around non-A2A expressing ? What about astrocytes? And what about afferents bearing A2AR from other brain regions?

Now looking the figure, they say that the frequency increases (not decreases), this should be clarified.

Figure 3A - Again local circuit vs projections: this makes more sense if CGS is affecting A2AR bearing projections onto LS (decreasing cfos), and possibly affecting LS-A2A+ activity of projections to LHb and DMH (increasing cfos)

Figure 3C - The quantification of neurons that are A2AR+ within the LS neurons is missing. Plus, if the virus used for anterograde and retrograde labeling is monosynaptic there is maybe room for distinct subpopulations?

Regardless of the low quality of the images, in the pictures showing DMH, the increase of cells labeled seems to be a lot more general, not restricted only to this DMH. It would be important to extend this analysis to other brain areas as well to claim the effects are specific.

If A2AR+ neurons are inhibitory and monosynaptically connected to LHb and DMH neurons, why is the net effect an increase in activity in the target structures? Not clear what is the author's interpretation of this finding.

Line 178 - No description whatsoever of protocols for opto-stimulation-light intensity pattern of stimulation, whether controls also were implanted

Figure 4 and 5: The low n of infected neurons (the best case shown has 3 GFP+ neurons in the slice) makes hard to understand the behavioral effect, it seems that only a very small subset of A2AR+ is actually responding to light. Furthermore, the axonal stimulation not necessarily implies selective activation of that specific connection. It is known that axonal spikes could backpropagate to the soma and/or other axonal branches. Thus, the overall effect of illumination could result in stimulation of all the postsynaptic targets (local and long-range) of LS A2AR+ neurons, making hard to claim that one of those pathways is solely underlying the effect found. It would be interesting to see if there is an additive effect of stimulating LSLHb and LSDMH at the same time. If that fails, that could mean that the stimulation of one pathway is also recruiting the other one.

Line 209, a trend is not visible in hippocampus.

Fig7E, Two of the three bands in hippocampus look decreased for 5d-RFFS.

Figure 8: Given that the AAV is cre-independent, the A2AR overexpression is not restricted to A2AR+ neurons, which could alter the whole LS circuitry. In fact, the infection seems to go far beyond LS boundaries, at least to the MS. Thus, it is hard to understand why the effects here recapitulate so closely those dissected specifically from LS A2AR+ neurons. The experiment would be more informative if restricting the overexpression to A2AR+ in LS using cre-dependent AAVs.

Figure 9. Again the infection is reaching MS, so not focal to LS. What is the interpretation of TST significantly less immobility time for shRNA mice?

DISCUSSION

Line 303 – This sentence is an overstatement since CGS infusion to LS could activate many A2AR in non LS-A2A+Gabaergic cells...A2A bearing afferents and astrocytes are an intrinsic part of the population of A2A+ cells/subcellular

Line 405 – There are missing details in which animals, how many, how was light regulated, what was the intensity required, pattern, time, etc

Line 408 - When infusing the drugs was the animal awake and restrained, freely moving, anesthetized? What were the procedures, for single and chronic infusion?

Reviewer #3 (Remarks to the Author):

The manuscript by Wang and colleagues examines the role of lateral septum adenosine A2A receptor (A2A-R)-expressing neurons in stress-induced depressive behaviours. The authors show an association between LS A2A-R signalling and depression-associated behaviour in mice. Using electrophysiological recordings in mouse brain slices, they found that stimulation of A2A-Rs increased the firing of LS-A2AR-

expressing neurons and suppressed the firing of neighbouring LS neurons. Optogenetic stimulation and inhibition of LS-A2AR neurons enhanced and attenuated depression-like behaviour, respectively. The authors also found that LS A2A-Rs were upregulated in mice following chronic stress and in LS tissue from human suicide completers. Overexpression of A2A-Rs in the LS induced depression-like behaviours, whereas disruption of A2A-R signalling attenuated depressive-like behaviors. Together these results suggest that targeting A2A-Rs and LS may have therapeutic potential in the treatment of depression. Finally, the authors used viral tracing and optogenetic stimulation of LS-A2A-R terminals to implicate projections to the lateral habenula (LHb) and dorsal medial hypothalamus (DMH) as key components of the circuit mediating the behavioural effects.

This is a novel study and the data generally appear to support the authors conclusions. However, since optogenetic stimulation antidromically activates LS A2AR+ neurons, the authors cannot rule out the possibility that LS A2AR+ collateral projections to brain regions other than LHb and DMH are responsible for the observed behavioural effects. Optogenetic inhibition of LHb and DMH terminals would be a more definitive experiment. Several methodological details are missing from the manuscript, including optogenetic stimulation protocols, which confound interpretation of the results. The statistical reporting also lacks sufficient detail. The value of statistical tests and exact p-values should be reported.

Major Comments

1) Optogenetic stimulation of LS A2AR+ terminals in the DMH and LHb also antidromically activates the LS A2AR+ neurons, thus the authors cannot rule out the possibility that DMH and LHb projecting LS A2AR+ neurons impact the behaviour via collateral axons projecting elsewhere. Photoinhibition of DMH and LHb terminals would be a more definitive test. It would be interesting to see if inhibition of LS A2AR+ terminals recapitulates the effects of LS A2AR+ neuronal photoinhibition or if the non-selective photoinhibition of the LS terminals has a similar effect to LS A2AR neuronal stimulation.

2) Circuit level analysis was performed using transsynaptic labeling, axonal tracing, and single-cell PCR. However, the proportion of LS projections to DMH and LHb that arise from LS A2AR+ versus non-LS A2AR neurons is unclear. A more informative approach would be to infect LS A2AR+ neurons with ChR2 and record light-evoked responses in LHb and DMH neurons.

3) For the optogenetic experiments, what equipment was used? What were the stimulation patterns? Was the behaviour examined with and without light stimulation?

4) Other methodological details needed include:

- the stock concentration of drugs and vehicle used in the acute slice experiments
- whether c-fos cell counting was performed blind to the treatment group
- the duration of cannula infusions and the timing of the infusion relative to the test
- the exact antibody used
- the mouse brain atlas used
- the filters and objectives used for imaging
- the mini analysis parameters

5) The values generated by statistic tests should be reported along with exact p-values, and degrees of freedom.

6) Figure 2A,B. The red fluorescence is difficult to see. Sample size and summary data are not provided for A2A – GAD65+67 colocalization data.

7) Figure 2C. Number of neurons is not indicated. Bands are not explained.

8) Figure 2E. How often was the washout successful? Summary data would be helpful. Were antagonists also perfused? The methods indicate antagonists were used in the ex vivo electrophysiology experiments.

9) Figure 8. Why are there 2 bands in some but not all A2AR lanes?

Minor

- Abbreviations should be defined at first occurrence.
- Figure 1a, the putative terminals are difficult to see in the representative LHb image. Is “putative terminals” or “axonal fibers” a more appropriate term for the EGPF fluorescence signal as it is unclear how terminals are distinguished from fibers?
- For the transsynaptic labelling experiment, Fig 1b; the font size of the experimental approach diagram is too small, the mouse line should be indicated (presumably the A2AR-cre mice were not used). A brief description of the approach would be helpful. This would also explain how viral serotypes used for transsynaptic tracing were still appropriate for opsin terminal experiment stimulation experiments.
- Line 541, what is “search” mode?
- Line 547, drug perfusion experiments are not referred to as chemogenetic experiments

Dear Editor,

First, we wish to thank you for the opportunity to resubmit a revised and ameliorated version of our manuscript #22-34938 entitled "Lateral septum adenosine A_{2A} receptors control stress-induced depressive-like behaviors via signaling to the hypothalamus and habenula". We extend our thanks to the Reviewers for their positive evaluation of our study and for their criticisms and suggestions to improve its contents and its presentation: their time and efforts are very warmly appreciated. We addressed all their criticisms and, guided by the suggestions, we performed additional experiments revised the manuscript accordingly. The detailed answers to each of the questions raised are tendered point-by-point, in order of appearance, as follows:

Reviewer #1:

The paper by M Wang and co-workers is entitled "Lateral septum adenosine A_{2A} receptors control stress-induced depressive-like behaviors via signaling to the hypothalamus and habenula".

The results obtained in this paper are extremely interesting and convincing.

The authors show that a subpopulation of lateral septum (LS) GABAergic adenosine A_{2A} receptors-positive neurons mediate depression-like effects in mice via projections to the dorsomedial hypothalamus (DMH) and to the lateral habenula (LHb).

They show that the optogenetic activation of LS adenosine A_{2A} receptor-positive neurons projection terminals to the DMH or to the LHb induced depressive-like behaviors in the tail suspension test (TST), without inducing locomotor effects or anxiety-like effects.

It is noteworthy that a selective upregulation of A_{2A} receptors in the LS was observed in two models of stress in mice and was also observed in postmortem human brains of suicide completers.

Finally, local injection in the LS of the selective A_{2A} receptor antagonist KW6002 and knockdown of A_{2A} receptors reduced immobility in the TST without affecting motor activity.

There are however some points that might be considered to improve the manuscript.

We thank for the Reviewer for his/her positive assessment of our study.

Introduction:

Classically, it is acknowledged that ventral tegmental area (VTA), medial prefrontal cortex (mPFC), lateral habenula (LHb), lateral septum (LS), amygdala and hippocampus are implicated in major depressive disorders as quoted by the authors (lines 111-113). Therefore, the reader would probably appreciate to read in the discussion section information about the connections of LS GABAergic adenosine A_{2A} receptors-positive neurons with these brain regions, especially mPFC and hippocampus. See PMID: 35443088 DOI: 10.1002/jnr.25052

We fully understand the Reviewer's concern and our drive when beginning this study was to find a connectivity between the LS and areas relevant for mood control such as the hippocampus, amygdala and prefrontocortical (PFC) areas. However, our data is clear in showing that there is no direct connection of LS-A_{2A}R⁺ GABAergic neurons with PFC, amygdala, VTA or hippocampus. To comply with the Reviewer's suggestion, we now inserted in the Discussion section the reverse perspective, i.e. that the PFC or hippocampus might modulate the activity of LS-A_{2A}R⁺ neurons (line 393 onwards), which is indeed an open question.

Does CGS21680 infusion into the LS induce an upregulation of c-Fos expression in these structures?

The administration of CGS21680 in the LS mostly increased c-Fos expression in the LHb and DMH, reflecting the tight and direct connection of the LS with both the LHb and DMH that was identified with the viral tracing experiments. In other brain areas associated with mood processing such as the PAG, VTA and amygdala, we did not observe any evident alteration of c-Fos expression upon administration of CGS21680 in the LS. These findings are explicitly presented in the text of the revised manuscript (lines 160-161) and are displayed in Supplementary Fig S2.

Results: Figure 5. Optogenetic activation of the projection terminals in the dorsomedial hypothalamus (DMN) induced a more robust increase in immobility in tail suspension test (TST) than in lateral habenula (LHb). This could be discussed more thoroughly.

We thank the Reviewer for this very wise suggestion. Our data show that the DMH receives a larger number of projections from LS-A_{2A}R⁺ neurons compared to the LHb, as shown in Figure 1A. This provides a reasonable difference for the more robust effect of stimulating the A_{2A}R-containing terminals in the DMN compared to the LHb. This is now inserted in the revised manuscript (lines 195-198).

Discussion:

The classical view is that metabotropic Glu2/3 receptor antagonists display antidepressant-like effects, not agonists.

So how can this fit with the sentence line 306-307?

After reading again the referenced study proposing the role of mGlu2/3R [1], we are now convinced that stronger evidence might be required to support how the putative mGlu2/3R agonist in the LS affects depressive-like symptoms. Therefore, in agreement with the Reviewer's questioning, we judged more prudent to delete the sentence from the revised manuscript (line 323 onwards).

Same paragraph: 5-HT1A agonists are potential antidepressants so the citation lines 311-312 and 326-328 is very interesting but the reference quoted is probably wrong.

We have corrected the description and choose the right reference [2] (line 326-328 and 344-346). Thanks for noting this mistake.

The hypothesis of a restoration of the activity of the HPA axis is indeed interesting and fits well with the data of Batalha (2013-2016). It is unknown whether local knockdown of A_{2A}R or KW6002 infused in LS impact corticosterone levels following stress and status of GC receptors.

We fully agree with the Reviewer's assessment. Indeed, we were inspired by the excellent data provided by Batalha and co-workers [3, 4] establishing a relationship between hippocampal A_{2A}R and the activity of the HPA axis. This drove us to test the hypothesis that LS A_{2A}R⁺ neurons being the potential bridge between hippocampus and DMH; we collected CSF samples after the behavioral tests upon local knockdown of A_{2A}R or infusion of KW6002 into the LS, in order to test corticosterone levels and the status of GC receptors. Unfortunately, the quality of the samples was not suitable for testing due to long postponing period during COVID-19 prevalence. To understand the mechanism of LS-A_{2A}R⁺→DMH circuit in the regulation of depressive-like symptoms, more work is on our to-do list, such as identifying the types of LS A_{2A}R⁺ neurons-projection cells in DMH and investigating how LS A_{2A}R⁺ neurons directly or indirectly influence the activity of HPA axis. We hope to be able to address the issue again in the future, but we believe that the interest of this working hypothesis justifies its presentation still as a hypothesis (as now explicitly designated) in the manuscript (lines 343-350).

In my opinion the sentences 357-361 are not very informative in the context of the study and might be substituted by a figure (see for example those drawn in the review by Patel. PMID: 35443088 DOI: 10.1002/jnr.25052).

It is really cool suggestion and we decided to provide more information in the form of Supplementary Table 2. Additionally, we deleted reference (64), which did not report an increased level of A_{2A}R in HD and added a reference to a new study on early PD (line 386).

Finally, in the current study only male mice were used. It is known that stress responses vary across sex and may underlie the heightened vulnerability to psychopathology in females. See:

PMID: 29316846 DOI: 10.1080/10253890.2017.1422488

PMID: 34404738 DOI: 10.1101/cshperspect.a039198

PMID: 28825715 DOI: 10.1038/nm.4386

PMID: 29548746 DOI: 10.1016/j.biopsych.2018.01.017

So, authors could investigate whether optogenetic suppression of LS A_{2A}R-positive neurons decrease the immobility time in the TST also in females.

We fully agree with the Reviewer's comment. Indeed, we have performed the optogenetic activation of LS A_{2A}R⁺ neurons to induce depression-like behaviors in female A_{2A}R-Cre mice. There was no difference between male and female group (Supplementary Figure S3). So, we used male mice for all other behavior tests. Of course, whether the change of A_{2A}R in female is same as male mice in animal models of depression, is still an interesting and important open question to investigate.

Reviewer #2:

The main idea of this manuscript of the circuit dissection is very good and the selected approach is very elegant. However, many important methodological details were overlooked (namely regarding photo-stimulation) and the results are over-interpreted as a consequence of some crucial controls being absent.

Major issues:

Methods: There is no explanation whatsoever of the photo-stimulation methods, which is key to validate the results from optogenetic manipulations. Without this information (light power, stimulation frequency, etc.) is not possible to evaluate these data properly.

of cells that are LS-A_{2A}R? is it a biological relevant enough number?

Appropriate description of controls in each of the experiments is missing, especially since all this is extremely minute. If LS-A_{2A}R+ neurons have their projections at the Lhb and DMH, why are the authors they measuring local effects in figure 2F?

CGS infused into LS will change cfos expression in target neurons in two ways:

- 1. direct A_{2A}R activation in LS-A_{2A}R+ neurons OR*
- 2. activation of A_{2A}R in afferent pathways of the LS, that then also affect their target regions....*

The infusion experiment does not allow any selective discrimination of the source of cell-mediate effects

The western blot is used to claim a selective LS overexpression of A_{2A}R in the stress model, however that is not supported by the figure

We very much thank the Reviewer for the detailed evaluation of our study, which compelled us to considerably rephrase and complete our manuscript in the hope of making its contents clearer, as well as to performed additional experiments to address several of the questions raised. Our responses are detailed below under each point raised by the Reviewer.

Please find below the detailed revision:

ABSTRACT:

Line 48 – the sentence of ‘depression being the single largest contributor...’ is not clear or objective. Should be something like ‘Depressive disorders rank as the #1 cause of disease burden worldwide’

We have replaced the sentence (line 42), which now reads ‘Major depressive disorder ranks as a major burden of disease worldwide’.

Line 49 – the claim of high non-responsiveness to anti-depressive treatments is not correct. Must be rephrased.

We have replaced the sentence (line 42-43), which now reads ‘the current antidepressant medications are limited by frequent non-responsiveness’.

Line 52 - Neurons and especially receptors do NOT mediate depression which is a disease with various symptoms...

We have replaced the “depression” with “depressive symptoms” (line 46).

Line 55 – It is not clear what: ‘formatted depressive phenotype’ means

We have eliminated this expression “formatted” (line 50).

Line 57 – Error: should be we have shown and NOT we shown

We rephrased the sentence to correct this error (line 52).

Line 59 – What is the meaning of ‘suicide completers’ ?

The term “suicide completers” is used for individuals who died committing suicide in contrast to “suicide attempters” or individuals with suicide ideation.

Line 117 – Please revise the sentence: ‘mediating depression via direct projects to...’

We have replaced the “depression” with “depressive symptoms” (line 110).

Line 126 – There is mention to a previous study but the reference is missing.

The reference is now included in the revised manuscript (line 119).

RESULTS:

Line 150 - These are non-labeled neurons presumably also GABAergic? Or not?

We presumed that all recorded non-labeled neurons are GABAergic due to two main reasons: first, previous studies have found that the LS is composed of predominantly GABAergic neurons (more than 90%) [5, 6]; in particular, the dorsal LS is reported to be made up of only inhibitory neurons [7]; second, our recording results also showed that all recorded non-labeled neurons display electrical properties characteristic of GABAergic neurons. This is now made explicit in the revised manuscript (lines 147-150).

Fig1A. The density of neurons seems too low, the quality of images is not good enough to see terminals, especially in LHb. A quantification of both, infected cells and terminals throughout different brain areas not restricted to the ones they show, could strengthen the data and give a real idea of the relative importance of these areas as LS output.

For clarity, we have replaced Figure 1A with a new version without DAPI straining. We found that

the LHb and DMH are the only two regions that receive projections from LS A_{2A}R⁺ neurons. No positive-EYFP signal was found in periaqueductal gray (PAG), ventral tegmental area (VTA) and raphe, which were reported in previous studies to receive projections from LS [8, 9].

Fig 1B. The quality of images should be improved, and quantifications could also help. There are areas of fluorescence outside the LH and DMH, for example in the dentate gyrus. This looks like the tract of the injection, but how could GFP be expressed in areas not targeted by the cre-aav infected in LS? If LS cre-expressing neurons target DG, it should be considered as a possible downstream effector when doing optogenetic experiments.

We performed the AAV-mediated anterograde transsynaptic tagging according to Zingg's work [10]: AAV1-Cre from transduced presynaptic neurons effectively and specifically drives Cre-dependent transgene expression in selected postsynaptic neuronal targets, thus allowing axonal tracing and functional manipulations of the input-defined neuronal population. Logically, the method cannot be used in A_{2A}R-Cre mice, which means our previous results only showed the direct connections between LS (not specifically LS-A_{2A}R⁺ neurons) and the LHb or DMH. Accordingly, considering the interconnection between septum and DG, it is not surprising to see the presence of fluorescence staining in DG in our figure. To avoid the misunderstanding, we deleted the Figure 1B. To better address this question, we now used a retrograde CTB488 tracing in A_{2A}R-tag mice to demonstrate the direct connection between LS-A_{2A}R⁺ neurons and the LHb or DMH (new Figure 1B-D), excluding the question of a projection into the DG.

Fig2A. Not clear which image areas are magnified in each magnification. What is the massive fluorescent label laterally to the ventricle, and outside the LS?

For clarity, we have replaced the figure 2A with a new version, which shows that a massive fluorescent is located in the striatum, where A_{2A}R is highly expressed. For *in vitro* recording, this intense fluorescent label was used to help us to find the location of the LS on the slices as A_{2A}R⁺ neurons are scarce in the LS.

Fig2B. Very hard to see GAD label, improving images quality could help. Also, this is a single example. It would be helpful to show immunos from slices with several GFP labeled neurons expressing GAD, and a quantification across slices/mice.

The figure is from a 300 µm slice after *in vitro* electrophysiology recording. The thickness of the slice made it difficult to clearly capture several EYFP-labeled neurons in the same focal plane. It is important to stress that we have performed an additional RNAscope experiment to confirm the co-location of A_{2A}R and GAD65+67 markers in the dorsal LS, which was carried out in 12 µm-thick slices. As we can see from the new figure (Supplementary Fig S1), all neurons endowed with A_{2A}R also expressed GAD65/67. These findings are fully aligned with a previous study reporting that the dorsal LS is made up of only inhibitory neurons [7].

Fig2C. Not clear if this is from a single cell or pooled from many cells, which could be confusing.

The line 2 panel displayed in Figure 2C corresponds to the analysis of a single neuron and this precise information is now provided in the description of the findings (lines 139-141) as well as in the legend to Figure 2 (line 9230-931). To make it further clear, we also modified the title of the methodological description corresponding to this analysis (line 633).

Figure 2F – The authors claim that ‘We found that A2AR activation by CGS21680 (30 nM) reduced the frequency (but

not amplitude) of spontaneous inhibitory postsynaptic currents (sIPSCs) of neurons around EYFP-positive cells, which likely resulted from the increase of A2AR+ GABAergic inputs to non-A2AR positive neurons in the LS (Figure 2F).

This claim is not clear, why would the frequency of inputs to A2AR- neurons decrease when neighboring GABAergic A2AR+ neurons are highly increasing their firing rate? If these neurons are connected to each other, as suggested by the authors, the frequency of sIPSCs should increase. This provided that A2AR+ neurons actually synapse onto A2AR- neurons which are not shown. Pair-recordings or optogenetic experiments are needed to show that this connection is real.

We are sorry for this inadvertent mistake and indebted to the Reviewer for its identification. In fact, we found that “A_{2A}R activation by CGS21680 (30 nM) increased the frequency (but not amplitude) of spontaneous inhibitory postsynaptic currents (sIPSCs) of neurons around EYFP-positive cells”. We have now corrected this error with “increased” in the Result section (line 150). This increase in sIPSCs by CGS21680 likely resulted from the increase of A_{2A}R⁺ GABAergic inputs to non-labeled neurons in the LS (Figure 2F), although this remains to be experimentally confirmed since indirect effects, circuit-mediated or involving astrocytes may eventually be also involved (lines 153-154). As wisely recommended by the Reviewer, one way to confirm if LS-A_{2A}R⁺ directly synapse onto LS-A_{2A}R⁻ neurons would be to carry out pair-recordings. However, all the work relative to alterations of neuronal activity in the LS essentially corresponds to a characterization of the relative role of A_{2A}R⁺ and A_{2A}R⁻ neurons in the LS, an effort that is certainly of great importance, but which was concluded not to be of prime importance for the control of mood and stress-related mood alterations. In fact, the optogenetic experiments of stimulation or inhibition of the LS-A_{2A}R⁺ terminals in the LHb and DMH clearly show that it is the projections of LS-A_{2A}R⁺ neurons to the LHb and DMH, rather than the impact of the activity of LS-A_{2A}R⁺ neurons within the LS circuitry, that play the key role controlling stress-induced mood alterations. This is now made explicit in the revised manuscript (lines 330-332). Thus, we detailed more the outer projections of LS-A_{2A}R⁺ neurons to the LHb and DMH while keeping in our-to-do list the clarification of the impact of the activity of LS-A_{2A}R⁺ neurons within the LS circuitry.

If you are postulating that A2A+ neurons project onto LHb and DMH, then how do you reconcile this with the local circuit effects they are claiming here to be due to the neurons that express A2A receptors onto neurons around non-A2A expressing? What about astrocytes? And what about afferents bearing A2AR from other brain regions?

As detailed in the response to the previous point, we now acknowledge in the revised manuscript that astrocytes may well play a role in the impact within the LS resulting from the selective activation of LS-A_{2A}R⁺ neurons (lines 153-154). The revised manuscript also explicit states that the observation that the optogenetic stimulation of the LS-A_{2A}R⁺ terminals in the LHb and DMH clearly show that it is the projections of LS-A_{2A}R⁺ neurons to the LHb and DMH, rather than the impact of the activity of LS-A_{2A}R⁺ neurons within the LS circuitry, that play the key role controlling stress-induced mood alterations (lines 3310-332). It will certainly be of interest to better understand if astrocytes are involved in the impact of the activity of LS-A_{2A}R⁺ neurons within the LS circuitry, but this should likely be attempted in a framework different from the current focus of understanding the role of LS-A_{2A}R in the control of stress-induced behavioral alterations.

Now looking the figure, they say that the frequency increases (not decreases), this should be clarified.

This has been corrected in the manuscript (line 150).

Figure 3A - Again local circuit vs projections: this makes more sense if CGS is affecting A2AR bearing projections onto

LS (decreasing cfos), and possibly affecting LS-A2AR+ activity of projections to LHb and DMH (increasing cfos)

We understand the rationale but, as mentioned in the previous two questions, our main focus was to attempt to clarify the role of LS-A_{2A}R⁺ neurons in the control of stress-induced behavioral alterations rather than the role of A_{2A}R in the control of intrinsic LS activity. The question raised by the Reviewer is certainly of great pertinence for further studies that we aim to carry out to better understand the role of A_{2A}R in the control of LS circuitry. To our knowledge, there is currently no evidence that LS receive A_{2A}R-bearing projections from other brain regions, which would be a tempting explanation to reconcile the ability of CGS21680 to increase neuronal responsiveness in the LS while decreasing global activity in the LS network. Therefore, we can neither confirm nor exclude the possibility that A_{2A}R-bearing afferents from the hippocampus, amygdala, prefrontal cortex, and entorhinal cortex, where low density A_{2A}R expression have been found [11, 12] and that may send projections to LS.

Figure 3C - The quantification of neurons that are A2AR+ within the LS neurons is missing. Plus, if the virus used for anterograde and retrograde labeling is monosynaptic there is maybe room for distinct subpopulations?

Following the Reviewer's suggestion, we now made it explicit in the revised manuscript the percent of A_{2A}R⁺ neurons within the LS (line 118). Retrograde labeling experiments using A_{2A}R-tag mice were performed. The labelled cells from LHb or DMH distribute in different sub-regions in LS (new Figure 1B-D). Thus, our results prompt the possibility that there might exist distinct subpopulations of A_{2A}R⁺ neurons in LS, which is certainly worth investigating in the future.

Regardless of the low quality of the images, in the pictures showing DMH, the increase of cells labeled seems to be a lot more general, not restricted only to this DMH. It would be important to extend this analysis to other brain areas as well to claim the effects are specific.

We agree with the Reviewer's point. We initially considered that the CGS21680 infusion into the LS would directly induce change of c-Fos expression in the brain regions that receive inputs from LS-A_{2A}R⁺ neurons. So, we only focused on DMH and LHb. Alterations found in other sub-regions in the hypothalamus may be an indirect effect considering the complex intrinsic circuitry of sub-regions of the hypothalamus.

If A2AR+ neurons are inhibitory and monosynaptically connected to LHb and DMH neurons, why is the net effect an increase in activity in the target structures? Not clear what is the author's interpretation of this finding.

To understand the potential mechanism, we proposed a disinhibition model, whereby LS-A_{2A}R⁺ neurons inhibit DMH or LHb GABAergic neurons leading to the disinhibition of DMH or LHb glutamatergic neurons. This is currently a hypothesis to interpret the data and much more work is required to address, although there are some arguments to support this hypothesis. In fact, the LHb is predominantly composed of glutamatergic neurons, whose dysregulation in the LHb contributes to depression [13], through alterations of glutamate levels, glutamate transport, GABA/glutamate ratio and function of GABA_A receptor, and AMPAR, CaMKII as well as NMDAR. Although whether the LHb contains GABAergic neurons is still a subject of debate, recent studies [14-16] that locally optogenetic activation of subtypes of GABAergic neurons in the LHb drives inhibitory responses in nearby cells, have indicated the existence of functional local inhibitory circuit within the LHb. Thus, we proposed the disinhibition model through a LS^{GABAergic} → LHb^{GABAergic} → LHb^{glutamatergic} circuit, which is now included in the Discussion of the revised manuscript (lines 367-368).

The dysregulation of the hypothalamus-pituitary-adrenal (HPA) axis is main cause for depression

development. Previous studies have identified kinds of neuropeptides signal involved in depression disorder, however, the role of inhibitory and excitatory neurons is largely unclear [17]. Recently, molecularly distinct populations of inhibitory and excitatory neurons have been found by single-cell transcriptomic analysis in the lateral hypothalamic area (LH) [18]. Furthermore, glutamatergic axons from the LH to the LHb were found to be engaged in aversion and stress-induced depression onset [19]. As for DMH, previous study has shown that the DMH expresses both vGLUT2 and vGAT, suggesting that the DMH is composed of both glutamatergic and GABAergic neurons [20]. To propose the disinhibition model through a $LS^{GABAergic} \rightarrow DMH^{GABAergic} \rightarrow DMH^{glutamatergic}$ circuit, more investigations are needed, like the role of DMH glutamatergic or GABAergic neurons in regulating depression, and the potential interaction between these different types of neurons.

Line 178 - No description whatsoever of protocols for opto-stimulation-light intensity pattern of stimulation, whether controls also were implanted

The information is now clearly stated the materials and methods section (lines 477-483). Specially, all control $A_{2A}R$ -Cre mice were implanted with 200 μ m diameter optic fibers after injection of AAV2/9-DIO-EYFP viruses.

Figure 4 and 5: The low n of infected neurons (the best case shown has 3 GFP+ neurons in the slice) makes hard to understand the behavioral effect, it seems that only a very small subset of $A_{2A}R+$ is actually responding to light.

We are sorry for the misunderstanding due to the low quality of the image, which has been replaced by a new version. Indeed, there are 9 neurons in the slice as we have shown. It is established that, under the bilaterally implanted optic fibers, there are about 30 neurons responding to light stimulation at the same time. The amazing power of $LS-A_{2A}R+$ neurons encouraged us to further study their discharge characteristics and structural characteristics (some data have shown in the manuscript Figure 1A). With respect to the apparently paradoxical relation between low number of $A_{2A}R$ and robust behavioral impact, we now stressed this point in the revised manuscript (lines 379-380).

Furthermore, the axonal stimulation not necessarily implies selective activation of that specific connection. It is known that axonal spikes could backpropagate to the soma and/or other axonal branches. Thus, the overall effect of illumination could result in stimulation of all the postsynaptic targets (local and long-range) of $LS A_{2A}R+$ neurons, making hard to claim that one of those pathways is solely underlying the effect found.

We appreciate the suggestion (the same one with the third reviewer; well, it seems that great minds think alike!) and performed the suggested experiment. We observed that the photoinhibition of DMH and LHb terminals from $LS A_{2A}R+$ neurons reduce the depression-like phenotypes (new Fig 5I-P), as described in the text of the revised manuscript (lines 199-209).

It would be interesting to see if there is an additive effect of stimulating $LS \rightarrow LHb$ and $LS \rightarrow DMH$ at the same time. If that fails, that could mean that the stimulation of one pathway is also recruiting the other one.

It is really interesting to consider the possible additive effect of stimulating $LS \rightarrow LHb$ and $LS \rightarrow DMH$ at the same time. However, LHb (-1.72 mm AP; \pm 0.46 mm ML; -2.7 mm DV) and DMH (-1.70 mm AP; \pm 0.5 mm ML; -5.0 mm DV) are close to each other. Additionally, only bilaterally stimulation of the LHb could induce change of behaviors in TST, which rules out the possibility of unilaterally stimulation of the DMH and LHb at the same time. In brief, the area does not seem to be large enough to implant 3-4 cannulas, straight or inclined, at the same time.

Line 209, a trend is not visible in hippocampus.

We agree with the Reviewer's comment and have changed the description in the text (lines 226-227).

Fig7E, Two of the three bands in hippocampus look decreased for 5d-RFFS.

We agree with the Reviewer's point. After performing the experiment again, we have replaced Figure 7E.

Figure 8: Given that the AAV is cre-independent, the A2AR overexpression is not restricted to A2AR+ neurons, which could alter the whole LS circuitry. In fact, the infection seems to go far beyond LS boundaries, at least to the MS. Thus, it is hard to understand why the effects here recapitulate so closely those dissected specifically from LS A2AR+ neurons.

The experiment would be more informative if restricting the overexpression to A2AR+ in LS using cre-dependent AAVs.

We are sorry for the misunderstanding due to the inappropriate position of the word "MS" in the half-brain figure. For clarity, we have replaced Figure 8A with its full-brain version showing the expression of A_{2A}R within the LS, not at the boundaries of the MS. We do agree with the Reviewer's contention that it is better to restrict the overexpression of A_{2A}R in the LS using cre-dependent AAVs in A_{2A}R-Cre mice. However, considering the great numbers of A_{2A}R-Cre mice that we have used for behaviors tests, we suppose ectopic expression of A_{2A}R in C57BL/6 mice is an acceptable alternative strategy.

Figure 9. Again the infection is reaching MS, so not focal to LS. What is the interpretation of TST significantly less immobility time for shRNA mice?

As the reason mentioned above, we have prepared a new Figure 9A.

DISCUSSION

Line 303 – This sentence is an overstatement since CGS infusion to LS could activate many A2AR in non LS-A2A+Gabaergic cells...A2A bearing afferents and astrocytes are an intrinsic part of the population of A2A+ cells/subcellular.

We appreciate the reviewer's critical comment on this. In fact, we got the conclusion from our slice recordings (see Fig 2) that the firing frequency of EYFP-positive neurons (that are A_{2A}R⁺ neurons) increased after CGS21680 (30 nM) infusion. The phenomenon may be a direct effect after activation of A_{2A}R on LS-A_{2A}R⁺ cells, or a combined direct and indirect effect after acting A_{2A}R in LS-A_{2A}R⁺ cells and A_{2A}R bearing afferents and astrocytes. Accordingly, we rephrased the sentence to encompass these two possibilities (line 153-154).

Line 405 – There are missing details in which animals, how many, how was light regulated, what was the intensity required, pattern, time, etc

This information is now clearly stated in the materials and methods section (lines 477-483).

Line 408 - When infusing the drugs was the animal awake and restrained, freely moving, anesthetized? What were the procedures, for single and chronic infusion?

This information is now clearly stated in the materials and methods section (lines 461-475).

Reviewer #3:

The manuscript by Wang and colleagues examines the role of lateral septum adenosine A2A receptor (A2A-R)-expressing neurons in stress-induced depressive behaviours. The authors show an association between LS A2A-R signalling and

depression-associated behaviour in mice. Using electrophysiological recordings in mouse brain slices, they found that stimulation of A_{2A}-Rs increased the firing of LS-A_{2A}R⁻ expressing neurons and suppressed the firing of neighbouring LS neurons. Optogenetic stimulation and inhibition of LS-A_{2A}R neurons enhanced and attenuated depression-like behaviour, respectively. The authors also found that LS A_{2A}-Rs were upregulated in mice following chronic stress and in LS tissue from human suicide completers. Overexpression of A_{2A}-Rs in the LS induced depression-like behaviours, whereas disruption of A_{2A}-R signalling attenuated depressive-like behaviors. Together these results suggest that targeting A_{2A}-Rs and LS may have therapeutic potential in the treatment

of depression. Finally, the authors used viral tracing and optogenetic stimulation of LS-A_{2A}-R terminals to implicate projections to the lateral habenula (LHb) and dorsal medial hypothalamus (DMH) as key components of the circuit mediating the behavioural effects.

This is a novel study and the data generally appear to support the authors' conclusions. However, since optogenetic stimulation antidromically activates LS A_{2A}R⁺ neurons, the authors cannot rule out the possibility that LS A_{2A}R⁺ collateral projections to brain regions other than LHb and DMH are responsible for the observed behavioural effects. Optogenetic inhibition of LHb and DMH terminals would be a more definitive experiment. Several methodological details are missing from the manuscript, including optogenetic stimulation protocols, which confound interpretation of the results. The statistical reporting also lacks sufficient detail. The value of statistical tests and exact p-values should be reported.

We thank the Reviewer for the helpful comments and detailed analysis. Following the comments, we performed additional experiments and addressed each question. Our responses are detailed below under each point raised by the Reviewer.

Major Comments

1) Optogenetic stimulation of LS A_{2A}R⁺ terminals in the DMH and LHb also antidromically activates the LS A_{2A}R⁺ neurons, thus the authors cannot rule out the possibility that DMH and LHb projecting LS A_{2A}R⁺ neurons impact the behaviour via collateral axons projecting elsewhere. Photoinhibition of DMH and LHb terminals would be a more definitive test. It would be interesting to see if inhibition of LS A_{2A}R⁺ terminals recapitulates the effects of LS A_{2A}R⁺ neuronal photoinhibition or if the non-selective photoinhibition of the LS terminals has a similar effect to LS A_{2A}R neuronal stimulation.

We appreciate the suggestion from the Reviewer and performed the experiment accordingly. We observed that the photoinhibition of DMH and LHb terminals from LS A_{2A}R⁺ neurons reduce the depression-like phenotypes (new Fig 5I-P), as described in the text of the revised manuscript (lines 199-209).

2) Circuit level analysis was performed using transsynaptic labeling, axonal tracing, and single-cell PCR. However, the proportion of LS projections to DMH and LHb that arise from LS A_{2A}R⁺ versus non-LS A_{2A}R neurons is unclear. A more informative approach would be to infect LS A_{2A}R⁺ neurons with ChR2 and record light-evoked responses in LHb and DMH neurons.

We very much appreciate the reviewer's critical comment on the importance in dissecting out the relative proportions of A_{2A}R⁻ neurons versus LS A_{2A}R⁺ (more importantly LS A_{2A}R⁺→DMH versus LS A_{2A}R⁺→LHb) for the LS A_{2A}R⁺ neuronal control of depression. We first addressed this issue of the relative proportions by injecting AAV2/9-DIO-EYFP into the LS of A_{2A}R-Cre mice and hope to identify sufficient numbers of the terminals in the DMH and LHb from LS-A_{2A}R⁺ neurons for electrophysiological recording. However, the weak and scarce fluorescent signals in the DMH or LHb of the specific brain sections that preserved the continuous and intact LS and DMH/LHb, made it very hard to obtain reliable electrophysiological recording despite multiple attempts. As the relative contributions of the LS

A_{2A}R⁺→DMH *versus* LS A_{2A}R⁺→LHb are critical to LS A_{2A}R⁺ control of depression, we employed our newly developed A_{2A}R-tag mice coupled with retrograde CTB488 approach to dissect out the relative proportion of LS-A_{2A}R⁺→DMH *versus* LS-A_{2A}R⁺→LHb. We injected CTB-488 unilaterally into DMH or LHb of A_{2A}R-tag mice, and determined the co-location of CTB and HA fluorescence in these brain regions. The results indicate that LS A_{2A}R⁺ neurons projected more abundantly to the DMH (25.2-29.2%) than the LHb (13.8%-21%) (Fig 2B-D). This provides the important insights into the LS A_{2A}R⁺ neuronal control of depression with the relative contribution of the LS-A_{2A}R⁺→DMH *versus* LS-A_{2A}R⁺→LHb projections. We have included this result and discussion the revised manuscript (line 124-128 and 195-198).

3) For the optogenetic experiments, what equipment was used? What were the stimulation patterns? Was the behaviour examined with and without light stimulation?

This information is now clearly stated in the materials and methods section (lines 477-483).

4) Other methodological details needed include:

- *the stock concentration of drugs and vehicle used in the acute slice experiments*

We have included the information in the text (line 600-627).

- *whether c-fos cell counting was performed blind to the treatment group*

Yes. We have added the statement in the text (lines 564).

- *the duration of cannula infusions and the timing of the infusion relative to the test*

We have included the information in the text (lines 462-475).

- *the exact antibody used*

This has been corrected throughout the manuscript.

- *the mouse brain atlas used*

We have included the information in the text (line 451-453).

- *the filters and objectives used for imaging*

This has been corrected throughout the manuscript.

- *the mini analysis parameters*

This has been corrected throughout the manuscript.

5) The values generated by statistic tests should be reported along with exact p-values, and degrees of freedom.

This has been corrected throughout the manuscript.

6) Figure 2A,B. The red fluorescence is difficult to see. Sample size and summary data are not provided for A2A – GAD65+67 colocalization data.

We do agree with the Reviewer and the Figure 2A is replaced by a new version. To address the fluorescence colocalization of A_{2A}R and GAD65+67 in LS, we have performed an RNAscope experiment. As we can see from the new inserted Figure, all labeled A_{2A}R neurons expressed GAD65+67 (Supplementary Figure S1).

7) Figure 2C. Number of neurons is not indicated. Bands are not explained.

The precise information is now provided in the description of the findings (lines 140-142) as well as in the legend to Figure 2 (line 930-931).

8) Figure 2E. How often was the washout successful? Summary data would be helpful. Were antagonists also perfused?

The methods indicate antagonists were used in the ex vivo electrophysiology experiments.

Following the Reviewer's suggestion, summary data is included in Fig 2E. The antagonist KW6002 was not used in the ex vivo electrophysiology experiments. We are sorry for the mistake, which we have corrected (line 627).

9) Figure 8. Why are there 2 bands in some but not all A2AR lanes?

We have done the experiments again and replaced a new picture in Fig.8B.

Minor

- Abbreviations should be defined at first occurrence.*

This has been corrected throughout the manuscript.

- Figure 1a, the putative terminals are difficult to see in the representative LHb image. Is "putative terminals" or "axonal fibers" a more appropriate term for the EGPF fluorescence signal as it is unclear how terminals are distinguished from fibers?*

We thank the reviewer for pointing the issue and we have replaced the Figure 1A. Following the Reviewer's suggestion, we preferentially used the terminology "axonal fibers" in the text (line 121).

- For the transsynaptic labelling experiment, Fig 1b; the font size of the experimental approach diagram is too small, the mouse line should be indicated (presumably the A2AR-cre mice were not used). A brief description of the approach would be helpful. This would also explain how viral serotypes used for transsynaptic tracing were still appropriate for opsin terminal experiment stimulation experiments.*

We performed the AAV-mediated anterograde transsynaptic tagging according to Zingg's work [10]: AAV1-Cre from transduced presynaptic neurons effectively and specifically drives Cre-dependent transgene expression in selected postsynaptic neuronal targets, thus allowing axonal tracing and functional manipulations of the latter input-defined neuronal population. Logically, the method cannot be used in Cre mice. Therefore, our results only indicate there are direct connections between the LS and the LHb (LS→LHb) or DMH (LS→DMH). Furthermore, we used retrograde CTB488 in A_{2A}R-tag mice to show the outputs from LS-A_{2A}R⁺ neurons to the LHb or DMH (new Fig 1B-D).

- Line 541, what is "search" mode?*

The sentence has been corrected (line 623).

- Line 547, drug perfusion experiments are not referred to as chemogenetic experiments*

The sentence has been corrected (line 627).

References

1. Wang, Y., et al., *mGlu2/3 receptors within the ventral part of the lateral septal nuclei modulate stress resilience and vulnerability in mice*. Brain Res, 2022. **1779**: p. 147783.
2. Kaster, M.P., A.R. Santos, and A.L. Rodrigues, *Involvement of 5-HT1A receptors in the antidepressant-like effect of adenosine in the mouse forced swimming test*. Brain Res Bull, 2005. **67**(1-2): p. 53-61.
3. Batalha, V.L., et al., *Adenosine A(2A) receptor blockade reverts hippocampal stress-induced deficits and restores corticosterone circadian oscillation*. Mol Psychiatry, 2013. **18**(3): p. 320-31.
4. Kaster, M.P., et al., *Caffeine acts through neuronal adenosine A2A receptors to prevent mood and memory dysfunction triggered by chronic stress*. Proc Natl Acad Sci U S A, 2015. **112**(25): p. 7833-8.
5. Risold, P.Y. and L.W. Swanson, *Chemoarchitecture of the rat lateral septal nucleus*. Brain Res Brain Res Rev, 1997. **24**(2-3): p. 91-113.
6. Zhao, C., B. Eisinger, and S.C. Gammie, *Characterization of GABAergic Neurons in the Mouse Lateral Septum: A Double Fluorescence In Situ Hybridization and Immunohistochemical Study Using Tyramide Signal Amplification*. PLoS ONE, 2013. **8**(8).
7. Besnard, A., et al., *Dorsolateral septum somatostatin interneurons gate mobility to calibrate context-specific behavioral fear responses*. Nat Neurosci, 2019. **22**(3): p. 436-446.
8. Sheehan, T.P., R.A. Chambers, and D.S. Russell, *Regulation of affect by the lateral septum: Implications for neuropsychiatry*. Brain Res Rev, 2004. **46**(1): p. 71-117.
9. Rizzi-Wise, C.A. and D.V. Wang, *Putting together pieces of the lateral septum: multifaceted functions and its neural pathways*. eNeuro, 2021. **8**(6).
10. Zingg, B., et al., *AAV-Mediated Anterograde Transsynaptic Tagging: Mapping Corticocollicular Input-Defined Neural Pathways for Defense Behaviors*. Neuron, 2017. **93**(1): p. 33-47.
11. Rosin, D.L., et al., *Immunohistochemical localization of adenosine A2A receptors in the rat central nervous system*. J Comp Neurol, 1998. **401**(2): p. 163-86.
12. Wang, M., et al., *Genetic tagging of the adenosine A2A receptor reveals its heterogeneous expression in brain regions*. Front Neuroanat, 2022. **16**: p. 978641.
13. Hu, H., Y. Cui, and Y. Yang, *Circuits and functions of the lateral habenula in health and in disease*. Nat Rev Neurosci, 2020. **21**(5): p. 277-295.
14. Flanigan, M.E., et al., *Orexin signaling in GABAergic lateral habenula neurons modulates aggressive behavior in male mice*. Nat Neurosci, 2020. **23**(5): p. 638-650.
15. Webster, J.F., et al., *Disentangling neuronal inhibition and inhibitory pathways in the lateral habenula*. Sci Rep, 2020. **10**(1): p. 8490.
16. Zhang, L., et al., *A GABAergic cell type in the lateral habenula links hypothalamic homeostatic and midbrain motivation circuits with sex steroid signaling*. Transl Psychiatry, 2018. **8**(1): p. 50.
17. Bao, A.M. and D.F. Swaab, *The human hypothalamus in mood disorders: The HPA axis in the center*. IBRO Rep, 2019. **6**: p. 45-53.
18. Mickelsen, L.E., et al., *Single-cell transcriptomic analysis of the lateral hypothalamic area reveals molecularly distinct populations of inhibitory and excitatory neurons*. Nat Neurosci, 2019. **22**(4): p. 642-656.
19. Zheng, Z., et al., *Hypothalamus-habenula potentiation encodes chronic stress experience and drives depression onset*. Neuron, 2022. **110**(8): p. 1400-1415 e6.
20. Xu, Y. and Q. Tong, *Expanding neurotransmitters in the hypothalamic neurocircuitry for energy balance regulation*. Protein Cell, 2011. **2**(10): p. 800-13.

We thank again the Reviewers for their comments and helpful suggestions, which contributed to improve our MS. We hope that, by addressing all the comments of the Reviewers, the revised version of the manuscript may prove acceptable for publication in *Nature Communications*.

Sincerely yours,

Wei Guo & Jiang-fan Chen

REVIEWERS' COMMENTS

Reviewer #1 (Remarks to the Author):

The authors have adequately answered all the questions I had formulated. I do not see any obstacle to the publication of the article.

Reviewer #2 (Remarks to the Author):

I have revised the paper and I think the authors did a pretty good job. The only major point that I see is still the low number of neurons (and their projections) expressing A2AR in the LS, to produce the behavioral phenotypes described. Data in figure 1D indicates that for 18 slices (3 mice x 6 slices), there are less than 2 A2AR+ neurons projecting to DMH and less than 1 A2AR+ neurons projecting to LHb per slice. If the counting was bilateral those numbers should be divided by 2 to get the number of neurons in a particular hemisphere, which seems very little compared to the behavioral output, but that would be up to the editor to judge.

I would say that the paper is more acceptable, although there are some points I would suggest the authors to address before publication.

1. In line 118 they say: Our previous study has shown that A2AR-positive (A2AR+) neurons make up about one percent of neurons in the LS, as shown by anti-HA staining in A2AR-tag mice [29]. I think authors should go back to this point in the discussion and propose how such a small number of neurons could have these behavioral effects.

2. In line 122 they refer to some "data not shown". Data should be shown for transparency.

3. In Results section 4, the authors show that A2ARs are upregulated in the LS in two mouse models of chronic stress. It would be interesting to see some interpretation regarding whether this is due to the same small population of A2AR+ neurons elevating their expression levels or more neurons starting to express A2AR. This would also impact results in figures 8 and 9 where A2A viral overexpression is not specific to A2AR+ neurons.

4. Line 247, authors should explain why sucrose consumption is an indicator of despair-like behavior.

5. Line 249, consider changing the word "trigger" for a less deterministic term.

6. In Figure 3, images in B seem to show that c-fos neurons are more abundant and more intense after CGS and not the opposite as stated in the text and quantification. In C, as noted in the first revision, the increase of cells labeled seems to be a lot more general, not restricted only to DMH. Although the authors replied this could be due to circuit effects, a quantification of neighboring DMH areas would be

informative to understand how specific the effect is.

7. In figure 4, is there any explanation for the difference in the O maze time in open arms for EYFP mice when expressing ChR2 (panel E, 300s) or eNpHR3 (panel J, 40s)?

Reviewer #3 (Remarks to the Author):

The authors have addressed most my concerns in the revised manuscripts. The suggested experiments were performed and strengthened the study. The results and methods are described in greater detail, although some additional details are still required for reproducibility. Specific comments are listed below.

For the optogenetic experiments, light was applied throughout the behavioral test. The authors should explain why they chose to conduct the experiment in this manner as it is generally more compelling to report the behavior before, during and after light stimulation.

The authors should clarify the patch recording mode used for the different electrophysiological parameters. Presumably some of the electrophysiology data were acquired using the whole-cell rather than the cell-attached patch-clamp configuration. The cell-attached configuration is not well-suited for recording IPSCs or filling the neuron with biocytin.

To enhance reproducibility the authors should specify:

- the version of software packages
- the detection parameters used in the mini analysis
- how the drugs applied were prepared (vehicle and stock concentration)
- setting used in fluorescence microscopy (objective, filters, etc)

Minor:

Lines 480-481. "0.0001 Hz, with 9999 s pulses" exceed the duration of the test.

Line 942 "n=7 cells, 4 mice, Paired t test, p=0.6927, W=21", there is a mismatch between the test and the test statistic.

Dear Editor,

We wish to thank you and the Reviewers for your helpful suggestions to further improve our MS. Guided by the suggestions, we revised the manuscript accordingly. The detailed answers to each of the questions raised are tendered point-by-point, as follows:

REVIEWERS' COMMENTS

Reviewer #1:

The authors have adequately answered all the questions I had formulated. I do not see any obstacle to the publication of the article.

We thank for the Reviewer's support.

Reviewer #2:

I have revised the paper and I think the authors did a pretty good job. The only major point that I see is still the low number of neurons (and their projections) expressing A2AR in the LS, to produce the behavioral phenotypes described. Data in figure 1D indicates that for 18 slices (3 mice x 6 slices), there are less than 2 A2AR+ neurons projecting to DMH and less than 1 A2AR+ neurons projecting to LHb per slice. If the counting was bilateral those numbers should be divided by 2 to get the number of neurons in a particular hemisphere, which seems very little compared to the behavioral output, but that would be up to the editor to judge.

We are sorry for the misunderstanding due to our description. In fact, the number (n=32 or 14) shown in the Fig 1D is the average, not total, number from 3 mice.

I would say that the paper is more acceptable, although there are some points I would suggest the authors to address before publication.

1. In line 118 they say: Our previous study has shown that A2AR-positive (A2AR+) neurons make up about one percent of neurons in the LS, as shown by anti-HA staining in A2AR-tag mice [29]. I think authors should go back to this point in the discussion and propose how such a small number of neurons could have these behavioral effects.

As we have discussed (lines 389-390) that the apparently paradoxical relation between the low number of A_{2A}R and robust behavioral impact may attribute to the discharge characteristics and structural characteristics (some data have shown in the manuscript Figure 2) of A_{2A}R⁺ neurons in LS. This idea is supported by previous findings that extra-striatal A_{2A}R have a low density but robustly affect brain functions. A_{2A}R blockade was reported to be efficient in reverting the behavioral and electrophysiological and morphological impairments induced by maternal separation [1]. Moreover, A_{2A}Rs are enriched in glutamatergic synapses in the amygdala, where they selectively control synaptic plasticity at a major afferent pathway to this brain region. Notably, the downregulation of A_{2A}Rs was shown to impair fear memory acquisition as well as fear memory retrieval in Pavlovian conditioning [2]. Of course, the characteristics of A_{2A}R⁺ neurons in LS underlying their robust behavioral effects require more and direct evidence.

2. In line 122 they refer to some "data not shown". Data should be shown for transparency.

We have added the data in Supplementary Figure 1.

3. In Results section 4, the authors show that A2ARs are upregulated in the LS in two mouse models of chronic stress. It would be interesting to see some interpretation regarding whether this is due to the same small population of A2AR+ neurons elevating their expression levels or more neurons starting to express A2AR. This would also impact results in figures 8 and 9 where A2A viral overexpression is not specific to A2AR+ neurons.

Outside striatum, the expression level of the A_{2A}R is very low, and only negative or weak A_{2A}R immunoreactivity could be detected, even in our A_{2A}R-tag mouse. So it is hard to identify the precise percent of A_{2A}R⁺ neurons in various brain regions, including LS. The possibility of more neurons starting to express A_{2A}R cannot be ruled out, as recent study reports that increased expression of A_{2A}R is found in neurons, astrocytes and microglia in lesional areas in specimen from Rasmussen's encephalitis patients [3]. However, in our previous study, we only found an increased density of A_{2A}R in glutamatergic terminals, not gliosomes, in the hippocampus after chronic stress [4]. So it is an interesting question to study the origin of the upregulated A_{2A}R in LS in the future.

4. Line 247, authors should explain why sucrose consumption is an indicator of despair-like behavior.

We have replaced the “despair” with “depressive” (line 260).

5. Line 249, consider changing the word "trigger" for a less deterministic term.

We have replaced the “trigger” with “modulator” (line 260).

6. In Figure 3, images in B seem to show that c-fos neurons are more abundant and more intense after CGS and not the opposite as stated in the text and quantification.

We thank the reviewer for pointing the mistake and we have replaced the Figure 3b.

In C, as noted in the first revision, the increase of cells labeled seems to be a lot more general, not restricted only to DMH. Although the authors replied this could be due to circuit effects, a quantification of neighboring DMH areas would be informative to understand how specific the effect is.

We have added the data in Supplementary Figure 3 and include the possible explanation (lines 166-169): Alterations found in other sub-regions in the hypothalamus may be an indirect effect considering the complex intrinsic circuitry of sub-regions of the hypothalamus.

7. In figure 4, is there any explanation for the difference in the O maze time in open arms for EYFP mice when expressing Chr2 (panel E, 300s) or eNpHR3 (panel J, 40s)?

We apologized for using percentage value rather than duration time in Figure 4j. We thank the reviewer for pointing the mistake and we have replaced the Figure 4j.

Reviewer #3:

The authors have addressed most my concerns in the revised manuscripts. The suggested experiments were performed and strengthen the study. The results and methods are described in greater detail, although some additional details are still required for reproducibility. Specific comments are listed below.

For the optogenetic experiments, light was applied throughout the behavioral test. The authors should explain why they chose to conduct the experiment in this manner as it is generally more compelling to report the behavior before, during and after light stimulation.

The protocol (performing behavior tests before, during and after light stimulation) has been widely used recently for behaviors. However, as tail-suspension for 6 min is believed to be a very strong stimulus for rodents [5, 6], to our limited knowledge, the above-mentioned protocol is still not widely adopted to study depression-like behaviors. To minimize the suffering of experimental mice, the protocol (light stimulation during behavior tests) was used in our study.

The authors should clarify the patch recording mode used for the different electrophysiological parameters. Presumably some of the electrophysiology data were acquired using the whole-cell rather than the cell-attached patch-clamp configuration. The cell-attached configuration is not well-suited for recording IPSCs or filling the neuron with biocytin.

We thank the reviewer for the comments. We clarified the recording configurations in our manuscript (lines 136-139): After cell-attached recording, we applied a negative pressure to form a gigaohm seal between the cell and the glass pipette, and then used a brief suction to break into the cell. Then the cell will be kept under whole cell voltage clamp mode for at least 10 minutes to allow the biocytin (1%, Thermo Scientific) in the intracellular solution to diffuse well into the cell.

To enhance reproducibility the authors should specify:

- *the version of software packages*

Softwares packages used as below, and clarified in method part (page 28)

Ocular 2.0 (image acquiring);

Clampfit 10.6 (Molecular Devices);

Mini Analysis Program 6.0.7 (Synaptosoft Inc., New Jersey)

- *the detection parameters used in the mini analysis*

More details regarding the detecting parameters were added (lines 639-641): Mini Analysis Program 6.0.7 (Synaptosoft Inc., New Jersey) was used to confirm the cell attached action potential firing rate after Clampfit analysis, with filtering value set to 100pA for action potential amplitude analysis.

- *how the drugs applied were prepared (vehicle and stock concentration)*

We used bath application (2-3mL/min) for all drugs used (dissolved in aCSF) (line 634).

- *setting used in fluorescence microscopy (objective, filters, etc)*

More details were added (lines 629-633) as the reviewer suggested: Patch clamp recordings were made from LS neurons under visualisation using an upright microscope (Olympus BX51XI) equipped with a 5x objective and a 40x water-immersion objective, infrared(IR) illumination, Nomarski optics, and an IR-sensitive video camera(Electro, Canada). A_{2A}R+ neurons with green fluorescence were visualized using a mercury lamp (Olympus, U-RFL-T) through a GFP filter on the microscope. Light or fluorescent imaging data were acquired using OCULAR software (version

2.0).

Minor:

Lines 480-481. "0.0001 Hz, with 9999 s pulses" exceed the duration of the test.

This parameter represents a continuous stimulation, not the total duration of the experiment. The duration of each experiment can be found in the methods.

Line 942 "n=7 cells, 4 mice, Paired t test, p=0.6927, W=21", there is a mismatch between the test and the test statistic.

This has been corrected in line 1004 and 1005.

References

1. Batalha, V.L., et al., *Adenosine A(2A) receptor blockade reverts hippocampal stress-induced deficits and restores corticosterone circadian oscillation*. *Mol Psychiatry*, 2013. **18**(3): p. 320-31.
2. Simoes, A.P., et al., *Adenosine A2A Receptors in the Amygdala Control Synaptic Plasticity and Contextual Fear Memory*. *Neuropsychopharmacology*, 2016. **41**(12): p. 2862-2871.
3. He, X.H., et al., *Upregulation of adenosine A2A receptor and downregulation of GLT1 is associated with neuronal cell death in Rasmussen's encephalitis*. *Brain Pathol*, 2020. **30**(2): p. 246-260.
4. Kaster, M.P., et al., *Caffeine acts through neuronal adenosine A2A receptors to prevent mood and memory dysfunction triggered by chronic stress*. *Proc Natl Acad Sci U S A*, 2015. **112**(25): p. 7833-8.
5. Gururajan, A., et al., *The future of rodent models in depression research*. *Nature Reviews Neuroscience*, 2019. **20**(11): p. 686-701.
6. Castagné, V., et al., *Rodent Models of Depression: Forced Swim and Tail Suspension Behavioral Despair Tests in Rats and Mice*. *Current Protocols in Neuroscience*, 2011. **55**(1).

We thank again the Reviewers and hope that the revised version of the manuscript proves acceptable for publication in *Nature Communications*.

Sincerely yours,

Wei Guo & Jiang-fan Chen